# Income determines the impact of cash transfers on HIV/AIDS: cohort study of 22.7 million Brazilians

Andréa F. Silva [1,2], Inês Dourado[1], Iracema Lua[1,2], Gabriela S. Jesus[1,3], Nathalia S. Guimarães[1], Gabriel A. S. Morais [1], Rodrigo V. R. Anderle[1], Julia M. Pescarini[2], Daiane B. Machado[2,4], Carlos A. S. T. Santos[2], Maria Y. Ichihara[2], Mauricio L. Barreto[1,2], Laio Magno [1,5], Luis E. Souza[1], James Macinko [6] & Davide Rasella [1,2,7] ✉

Living with extremely low-income is an important risk factor for HIV/AIDS and can be mitigated by conditional cash transfers. Using a cohort of 22.7 million low-income individuals during 9 years, we evaluated the effects of the world's largest conditional cash transfer, the *Programa Bolsa Família*, on HIV/AIDS-related outcomes. Exposure to *Programa Bolsa Família* was associated with reduced AIDS incidence by 41% (RR:0.59; 95%CI:0.57-0.61), mortality by 39% (RR:0.61; 95%CI:0.57-0.64), and case fatality rates by 25% (RR:0.75; 95%CI:0.66-0.85) in the cohort, and *Programa Bolsa Família* effects were considerably stronger among individuals of extremely low-income [reduction of 55% for incidence (RR:0.45, 95% CI:0.42-0.47), 54% mortality (RR:0.46, 95% CI:0.42-0.49), and 37% case-fatality (RR:0.63, 95% CI:0.51 −0.76)], decreasing gradually until having no effect in individuals with higher incomes. Similar effects were observed on HIV notification. *Programa Bolsa Família* impact was also stronger among women and adolescents. Several sensitivity and triangulation analyses demonstrated the robustness of the results. Conditional cash transfers can significantly reduce AIDS morbidity and mortality in extremely vulnerable populations and should be considered an essential intervention to achieve AIDS-related sustainable development goals by 2030.

Living with extremely low-income is a well-recognized risk factor for a wide range of diseases and health conditions, including HIV/AIDS[1,2], which is responsible for a pandemic that caused more than 34 million deaths worldwide[3]. There is a growing consensus that HIV/AIDS control interventions should be focused not only on healthcare, but also on Social Determinants of Health (SDH), which have the potential to

reduce both morbidity and mortality related to HIV/AIDS[1,2,4]. Several studies have indicated that living with low-income can contribute to the occurrence of new cases of HIV/AIDS, although wealth may also play a role in driving HIV transmission among certain populations[5]. Individuals with socioeconomic vulnerabilities are at a higher risk not only of acquiring HIV but also of encountering obstacles in accessing

[1]Institute of Collective Health, Federal University of Bahia (UFBA), Salvador, Brazil. [2]Center for Data and Knowledge Integration for Health (CIDACS), Gonçalo Moniz Institute, Oswaldo Cruz Foundation (FIOCRUZ), Salvador, Brazil. [3]Faculty of Medicine, Federal University of Bahia (UFBA), Salvador, Brazil. [4]Department of Global Health and Social Medicine, Harvard Medical School, Boston, MA, USA. [5]Department of Life Sciences, State University of Bahia (UNEB), Salvador, Brazil. [6]Departments of Health Policy and Management and Community Health Sciences, UCLA Fielding School of Public Health, Los Angeles, CA, USA. [7]ISGlobal, Hospital Clinic - Universitat de Barcelona, Barcelona, Spain. ✉e-mail: davide.rasella@gmail.com

appropriate, timely, and continuous care and treatment. To effectively reduce the morbidity and mortality associated with AIDS, interventions need to address not only healthcare but also the SDH[6]. Moreover, HIV/AIDS reinforces the cycle of low-income perpetuation due to its frequent association with stigma, which can further exacerbate extremely low-income among populations already living in socially vulnerable circumstances[7].

Conditional cash transfer (CCT) programs are among the most effective and widespread interventions acting on SDH and individuals living with low and extremely low-income, and they have been implemented in almost all low- and middle-income countries (LMICs) to improve the well-being of families living in precarious conditions[8]. CCT programs transfer cash to low-income households with the requirement that parents comply with specific conditions (or conditionalities), usually focused on health and education for their children. CCT programs have demonstrated positive effects on the use of preventive services, the promotion of healthy behavior, and the improvement of a wide range of health outcomes[8]. However, available research findings are still inconclusive for HIV/AIDS:[4] while some studies have shown the effects of CCTs reducing the incidence and prevalence of HIV[9–13], as well as mitigating mother-to-child HIV transmission[14], many others have found no significant impacts[15].

In the last two decades, Brazil has implemented one of the world's largest and more extensively evaluated CCTs, the *Programa Bolsa Família* (PBF)[16]. The objective of the PBF was to alleviate poverty by providing cash transfers along with educational and health-related conditions. Implemented in 2004, the program achieved nationwide coverage, with enrollment of 14.2 million families by 2018. It involved direct cash transfers from the government to low-income households, defined as families earning between US$18-36 per person per month in 2018 (at an exchange rate of 5 Brazilian real to 1 US dollar). The amount of the monthly cash benefits ranged from $17 to a maximum of $41, depending on household size and composition, with funds being deposited onto a beneficiary debit card. To continue receiving the benefits, beneficiaries were required to meet certain conditions related to healthcare and education for their children. For instance, pregnant women were expected to attend antenatal and postnatal consultations, while children were required to receive regular nutrition monitoring and vaccinations. School-aged children had to attend school. Although the program primarily focused on families with children, adult individuals living in low-income or extremely low-income without children could also qualify for and receive the benefits[16]. The large-scale and rapid increase of PBF coverage has been associated with strong nationwide reductions in poverty and social inequalities, and the improvement of several health outcomes and conditions[11–13].

The success of the Brazilian response to HIV/AIDS is recognized worldwide, especially for its universalization of antiretroviral therapy (ART) in the 1990s and the current distribution of free pre-exposure prophylaxis[17]. However, the incidence of AIDS remains relatively high and unequally distributed in society[18], with the current national rate standing at 16.5 cases per 100,000 inhabitants in 2021[19].

To assess eligibility for social welfare programs (including PBF), Brazil has developed a large longitudinal administrative dataset corresponding to the lower-income half of its population that, linked to all nationwide AIDS-related records, has provided a unique opportunity to study the effects of CCTs on AIDS outcomes in the most vulnerable individuals – who are not usually included in epidemiological studies or randomized controlled trials.

Therefore, the aim of our study was to comprehensively evaluate the impact of CCTs on sequential AIDS outcomes – namely incidence, mortality, and case-fatality rate – using a cohort of 22.7 million Brazilians between 2007 and 2015.

## Results

### The 100 Million Brazilian Cohort, the linkages, and the selection process

The selection process to achieve our final cohort, which was composed of 22,788,998 individuals between 2007 and 2015 (the period for which HIV/AIDS data were available) is described in Fig. 1 and detailed in the appendix p.3. Among the 22,788,998 individuals of the cohort, 57,99% (13,214,290) were female and 42,01% (9,574,707) male. The population under study was achieved by selecting all individuals aged 13 and older from the 100 Million Brazilian Cohort[20], a consolidated cohort created through the linkage between the Federal Government Unified Registry for Social Programs (*Cadastro Único*) – that gathers data from the lower-income half of the Brazilian population, identifying and characterizing low-income families for social programs eligibility - and health-related datasets from the Brazilian Ministry of Health's (appendix, p.3)[20,21]. Two individual-level health-related datasets were linked to the Unified Registry for Social Programs: the Notifiable Diseases Information System (SINAN) and the Mortality Information System (SIM)[22]. SINAN is a decentralized information system that monitors the incidence of notifiable diseases, including HIV/AIDS[22]. SIM is a national death surveillance system that registers deaths by all causes, including HIV/AIDS, according to CID-10 classification. The quality of each link between Unified Registry, SINAN e SIM has been extensively evaluated and validated[20,23]. An aggregated dataset - containing municipal-level information on AIDS endemicity levels, municipal infrastructures, and healthcare resources - was also deterministically linked to the Unified Registry through the individuals' municipal code of residence and baseline year.

During the cohort follow-up, 22,212 new AIDS cases were detected (Table 1): 9201 cases occurred among PBF recipients, and 13,011 among non-recipients. There were 7650 deaths from AIDS: 42.2% occurred in PBF beneficiary individuals, and 57.8% in non-beneficiaries. Among PBF non-beneficiary individuals, the AIDS incidence rate over the study period was higher (29.7/100,000 person-years) than among beneficiaries (24.9/100,000 person-years). Comparable results were found for AIDS mortality (10.1/100,000 person-years for non-beneficiary individuals and 8.7/100,000 person-years for beneficiary individuals) and case-fatality (9.3/100 person-years for non-beneficiary individuals and 6.9/100 person-years for beneficiary individuals) rates, which were higher among non-beneficiaries.

The number of AIDS cases and AIDS incidence in the study cohort (22,212) is compatible with the AIDS incidence rates in Brazil (the small difference can be explained by the age and socioeconomic composition of the cohort), as shown in Table 1. The appendix contains additional analysis (Table S31, p. 46) that demonstrates the behavior of covariates between the beneficiaries and non-beneficiaries groups after applying IPTW weighting.

### The IPTW multivariable regression analyses

The adjusted Rate Ratios (RR) for the associations in the total population under study, estimated using multivariable Poisson regressions and weighted by the Inverse Probability of Treatment Weight (IPTW), are presented in Table 2 (for details on the adjusted models, see the methods section). The multivariable logistic regression models used to predict the propensity scores and, consequently, the IPTW, including the odds ratios (OR) for each set of confounding variables, and each outcome, are shown in Table S10 in the appendix. As presented in Table 2, the receipt of PBF benefits was associated with a 41% reduction in the AIDS incidence rate (RR:0.59; 95%CI:0.57−0.61), in comparison with non-beneficiaries. Receiving PBF was also associated with a 39% reduction (RR:0.61; 95%CI:0.57−0.64) in the AIDS mortality rate and a 25% reduction (RR:0.75; 95%CI: 0.66−0.85) in the case-fatality rate. Lastly, the socioeconomic and demographic covariates showed an overall, protective effect of being female, white, having higher

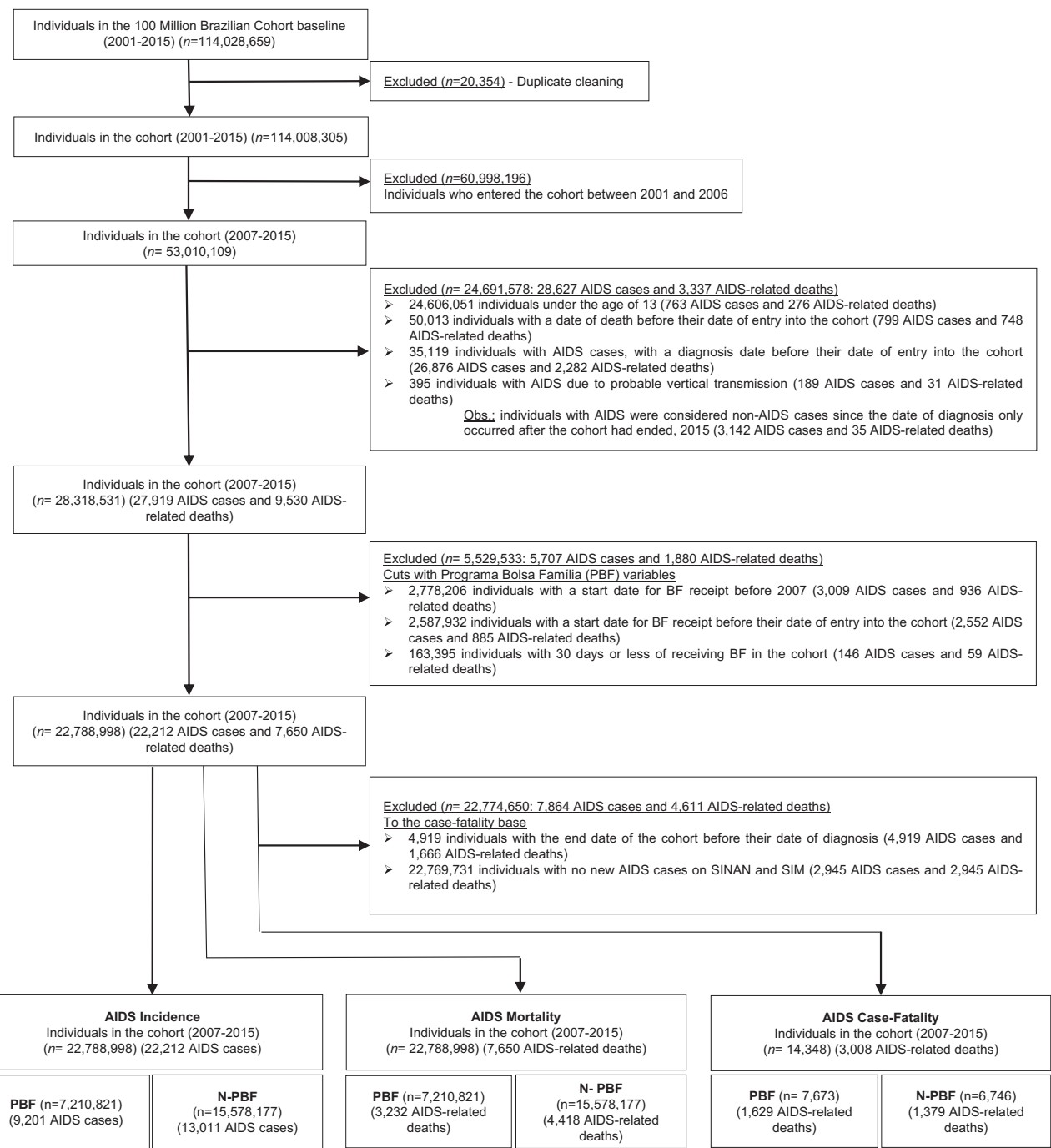

**Fig. 1 | Flowchart of the construction of the study cohort (Brazil, 2007–2015).**

education, and having higher wealth. On the other hand, a greater risk was found among individuals who lived in urban areas, in houses with precarious materials and infrastructure (water supply and electric power), black, adults, and never attended school.

## The PBF effect according to wealth levels, sex, and age

In the stratified analyses (Table 3), the associations between the receipt of PBF benefits and all AIDS indicators were considerably stronger among the extremely low-income individuals (in the 1st quartile of the distribution of the wealth variable), with statistically significant reductions of 55% in AIDS incidence (RR:0.45; 95%CI: 0.42-0.47), 54% in AIDS mortality (RR:0.46; 95%CI: 0.42-0.49), and 37% in AIDS case-fatality (RR:0.63; 95%CI: 0.51-0.76) rates. These reductions showed a consistent gradient over the levels of wealth, that is, PBF effects were

progressively lower as far as the individuals have higher levels of income: for quartiles 2 (37% in AIDS incidence, 35% in AIDS mortality, and 23% in AIDS case-fatality rates) and 3 (16% in AIDS incidence, 16% in AIDS mortality, and no effect on AIDS case-fatality rates). In the 4th quartile, representing the individuals with the highest income of the study cohort (defined empirically as individuals with capita expenses - proportional to the baseline minimum wage (MW) – above 56.5%) and a negative control for our analyses[24], PBF had no statistically significant effects. Receiving PBF benefits was also more strongly associated with a reduction in AIDS morbidity and mortality in women (of 40% in AIDS incidence, 42% in AIDS mortality, and 35% in AIDS case-fatality rates) and adolescents (of 52% in AIDS incidence, 54% in AIDS mortality).

The complete models with all covariates shown in Table 3 can be found in the appendix, pp.44–54.

**Table 1 | Descriptive analyses of *Programa Bolsa Família* (PBF) beneficiaries (BF) and non-beneficiaries (N-BF) (*n* = 22,788,998), 2007-2015**

| AIDS indicators | | N-BF (*n* = 15,578,177) | | BF (*n* = 7,210,821) | | Cohort Rates | Brazil Rates |
|---|---|---|---|---|---|---|---|
| | | Rates | No. | Rates | No. | | |
| Incidence Rate | | 29.68 | 13,011 | 24.88 | 9,201 | 27.4 | 20.2 |
| Mortality Rate | | 10.08 | 4,418 | 8.73 | 3,232 | 9.4 | 6.2 |
| Case-Fatality Rate | | 9.33 | 1,379 | 6.96 | 1,629 | - | - |
| Social and Demographic Variables | | N-BF (*n* = 15,578,177) | | BF (*n* = 7,210,821) | | *P* value[a] | SMD[b] |
| | | No. or Mean | (%) or CI | No. or Mean | (%) or CI | | |
| Sex | Female | 8,707,472 | 55.9 | 4,506,818 | 62.5 | <0.001 | 0.1347 |
| | Male | 6,870,704 | 44.1 | 2,704,003 | 37.5 | | |
| Age | Adolescents and young people[c] | 3,726,930 | 23.9 | 2,365,801 | 32.8 | <0.001 | 0.4719 |
| | Adults[d] | 9,876,398 | 63.4 | 4,746,382 | 65.8 | | |
| | Older people[e] | 1,974,296 | 12.7 | 98,355 | 1.4 | | |
| Race/ethnicity | White | 5,647,286 | 38.3 | 2,139,779 | 30.9 | <0.001 | 0.1983 |
| | Mixed-race | 8,020,979 | 54.4 | 4,083,887 | 58.9 | | |
| | Black | 1,058,718 | 7.2 | 617,395 | 8.9 | | |
| | Indigenous | 30,641 | 0.2 | 89,731 | 1.3 | | |
| Education | More than high school | 468,239 | 3.2 | 65,484 | 1.0 | <0.001 | 0.2227 |
| | High school | 4,299,384 | 29.7 | 1,854,519 | 28.0 | | |
| | Elementary school | 7,965,328 | 55.0 | 4,141,411 | 62.6 | | |
| | Attended pre-school | 163,856 | 1.1 | 51,134 | 0.8 | | |
| | Never attended school | 1,597,706 | 11.0 | 499,37 | 7.6 | | |
| Wealth[f] | Level 1 (More wealth) | 1,778,143 | 11.4 | 128,143 | 1.8 | <0.001 | 0.6983 |
| | Level 2 | 3,985,174 | 25.6 | 653,054 | 9.1 | | |
| | Level 3 | 2,624,084 | 16.8 | 1,127,457 | 15.6 | | |
| | Level 4 | 4,104,278 | 26.4 | 3,377,443 | 46.8 | | |
| | Level 5 (Lower wealth) | 3,081,951 | 19.8 | 1,923,399 | 26.7 | | |
| Water supply | Public network | 12,030,056 | 79.6 | 5,247,999 | 74.8 | <0.001 | 0.1136 |
| | Other[g] | 3,081,860 | 20.4 | 1,763,448 | 25.2 | | |
| Housing materials (brick) | Yes | 12,716,001 | 84.1 | 5,732,902 | 81.8 | <0.001 | 0.0634 |
| | No[h] | 2,395,729 | 15.9 | 1,278,437 | 18.2 | | |
| Lighting | Electricity | 13,896,551 | 92.0 | 5,953,494 | 84.9 | <0.001 | 0.2216 |
| | Non-electric[i] | 1,215,290 | 8.0 | 1,057,917 | 15.1 | | |
| Region | North | 1,469,601 | 9.4 | 882,9 | 12.2 | <0.001 | 0.5329 |
| | Northeast | 4,649,602 | 29.8 | 2,146,320 | 29.8 | | |
| | Southeast | 5,801,579 | 37.2 | 3,094,136 | 42.9 | | |
| | South | 2,168,128 | 13.9 | 592,030 | 8.2 | | |
| | Central-west | 1,488,877 | 9.6 | 495,399 | 6.9 | | |
| Area of residence | Rural | 2,702,484 | 17.4 | 1,284,501 | 17.9 | <0.001 | 0.0128 |
| | Urban | 12,836,718 | 82.6 | 5,899,174 | 82.1 | | |
| Average AIDS incidence rate[j] | | 19.14 | (19.13–19.14) | 21.66 | (21.65–21.67) | <0.001 | −0.1793 |
| Average AIDS mortality rate[j] | | 5.632 | (5.62–5.63) | 6.452 | (6.44–6.45) | <0.001 | −0.1865 |
| Average AIDS case-fatality rate[j] | | 30.88 | (30.87–30.89) | 30.86 | (30.85–30.86) | <0.001 | 0.0029 |
| Inadequate sanitation[k] | | 10.01 | (10.00–10.02) | 10.89 | (10.88–10.90) | <0.001 | −0.0688 |
| Unemployment rate (%)[l] | | 7.38 | (7.38–7.38) | 8.46 | (8.45–8.46) | <0.001 | −0.2892 |
| Doctors per 1,000 inhabitants[m] | | 1.34 | (1.34–1.35) | 1.46 | (1.46–1.47) | <0.001 | −0.1121 |
| Nurses per 1,000 inhabitants[m] | | 0.69 | (0.69–0.69) | 0.67 | (0.67–0.67) | <0.001 | 0.0480 |
| Hospital beds per 1,000 inhabitants[m] | | 2.32 | (2.32–2.32) | 2.33 | (2.33–2.33) | <0.001 | −0.0070 |

[a]The following were used for a comparison between the groups: (i) the two-tailed t-test for continuous variables and (ii) the Pearson's chi-squared test ($\chi^2$) for categorical variables.
[b]*SMD* Standardized mean difference.
[c]Aged between 13 and 24.
[d]Aged between 25 and 64.
[e]Aged 65 or older.
[f]Measured by capita expenses proportional to the baseline minimum wage (MW). Level 1 (More wealth): "1 or more". Level 2: "0.5 to 1". Level 3: "0.25 to 0.49". Level 4: "0< to 0.24".Level 5 (Lower wealth): "Nothing declared."
[g]Water supply: Other – well, spring, and others.
[h]Housing Material: No – Coated clay, uncoated clay, wood, and others.
[i]Lighting: Non-electric – No meter, lamps, candles, and others.
[j]Average rates for the period (2007–2015) by municipality.
[k]% of the municipal population with inadequate baseline sanitation.
[l]Baseline municipal unemployment rate.
[m]Per 1,000 inhabitants of the baseline municipality. All statistical tests used where two-sided.

**Table 2 | Estimates of the average effect of the *Programa Bolsa Família* (PBF) adjusted IPTW Poisson model (with robust standard errors) on AIDS incidence, mortality, and the case-fatality rates, 2007-2015**

| Adjusted Model | | Outcomes (RR[a] – CI[b] 95%) | | |
|---|---|---|---|---|
| | | Incidence | Mortality | Case-Fatality |
| *Programa Bolsa Família* | | 0.59*** (0.57–0.61) | 0.61*** (0.57–0.64) | 0.75*** (0.66–0.85) |
| Sex | Female | 1 (base) | 1 (base) | 1 (base) |
| | Male | 1.21*** (1.17–1.25) | 1.35*** (1.27–1.42) | 1.27*** (1.13–1.43) |
| Age | Adolescents and young people[c] | 1 (base) | 1 (base) | 1 (base) |
| | Adults[d] | 2.02*** (1.93–2.10) | 3.50*** (3.20–3.82) | 1.69*** (1.40–2.04) |
| | Older people[e] | 0.32*** (0.28–0.37) | 0.78* (0.64–0.96) | 2.84*** (1.78–4.54) |
| Race/ethnicity | White | 1 (base) | 1 (base) | 1 (base) |
| | Mixed-race | 1.24*** (1.19–1.29) | 1.24*** (1.15–1.32) | 1.07 (0.92–1.25) |
| | Black | 1.66*** (1.57–1.75) | 1.78*** (1.64–1.94) | 1.01 (0.83–1.22) |
| | Indigenous | 1.36* (1.00–1.85) | 0.89 (0.49–1.62) | 0.57 (0.20–1.61) |
| Education | More than high school | 1 (base) | 1 (base) | 1 (base) |
| | High school | 1.04 (0.91–1.18) | 1.45** (1.07–1.94) | 1.95* (1.06–3.58) |
| | Elementary school | 1.40*** (1.23–1.60) | 2.61*** (1.96–3.49) | 2.77** (1.53–5.02) |
| | Attended pre-school | 0.91 (0.72–1.15) | 1.20 (0.75–1.91) | 1.00 (0.35–2.78) |
| | Never attended school | 1.28*** (1.11–1.47) | 2.74*** (2.03–3.70) | 3.62*** (1.95–6.74) |
| Wealth[f] | Level 1 (More wealth) | 1 (base) | 1 (base) | 1 (base) |
| | Level 2 | 0.98 (0.89–1.08) | 0.91 (076–1.08) | 1.22 (0.80–1.87) |
| | Level 3 | 1.16* (1.05–1.28) | 1.22* (1.03–1.45) | 1.61* (1.06–2.43) |
| | Level 4 | 1.57*** (1.42–1.73) | 1.55*** (1.31–1.84) | 1.31 (0.87–1.98) |
| | Level 5 (Lower wealth) | 2.13*** (1.93–2.36) | 2.33*** (1.95–2.79) | 1.56* (1.02–2.40) |
| AIDS treatment | Yes | x | x | 1 (base) |
| | No | x | x | 2.67*** (2.32–3.08) |
| Water supply | Public network | 1 (base) | 1 (base) | 1 (base) |
| | Other[g] | 1.01 (0.96–1.06) | 1.07 (0.98–1.16) | 1.03 (0.87–1.23) |
| Housing material (brick) | Yes | 1 (base) | 1 (base) | 1 (base) |
| | No [h] | 1.24*** (1.19–1.30) | 1.33*** (1.24–1.44) | 1.18* (1.00–1.39) |
| Lighting | Electricity | 1 (base) | 1 (base) | 1 (base) |
| | Non-electric [i] | 1.31*** (1.25–1.38) | 1.37*** (1.26–1.48) | 1.21* (1.02–1.43) |
| Region | North | 1 (base) | 1 (base) | 1 (base) |
| | Northeast | 1.19*** (1.11–1.26) | 1.03 (0.92–1.16) | 1.06 (0.81–1.37) |
| | Southeast | 1.19*** (1.11–1.27) | 1.38*** (1.22–1.55) | 1.10 (0.83–1.45) |
| | South | 1.52*** (1.40–1.65) | 1.40*** (1.22–1.62) | 0.95 (0.70–1.30) |
| | Central-west | 1.22*** (1.12–1.34) | 1.19* (1.02–1.40) | 1.20 (0.85–1.68) |
| Area of residence | Rural | 1 (base) | 1 (base) | 1 (base) |
| | Urban | 1.89*** (1.77–2.03) | 2.21*** (1.97–2.49) | 1.06 (0.79–1.42) |
| Mun. average AIDS incidence rate[j] | | 1.03*** (1.02–1.03) | x | x |
| Mun. average AIDS mortality rate[j] | | x | 1.10*** (1.09–1.10) | x |
| Mun. average AIDS case-fatality rate[j] | | x | x | 1.01*** (1.00–1.02) |
| Inadequate sanitation[k] | | 0.99*** (0.99–0.99) | 0.99*** (0.98–0.99) | 1.00 (0.99–1.01) |
| Unemployment rate (%)[l] | | 1.01*** (1.01–1.02) | 1.02*** (1.01–1.03) | 0.98 (0.96–1.01) |
| Doctors per 1,000 inhabitants[m] | | 1.02 (0.99–1.05) | 1.05* (1.00–1.10) | 0.85** (0.76–0.95) |
| Nurses per 1,000 inhabitants[m] | | 0.85*** (0.78–0.92) | 0.86 (0.74–1.00) | 1.71** (1.23–2.36) |
| Hospital beds per 1,000 inhabitants[m] | | 0.99 (0.98–1.01) | 0.99 (0.99–1.00) | 0.97 (0.92–1.01) |
| Individual's year of entry into the cohort | | yes | yes | yes |
| Obs.: | | 19,577,629 | 19,577,649 | 9,965 |

***p value < 0.001; **p value < 0.01; *p value < 0.05.
[a]Incidence Rate Ratios.
[b]Confidence interval.
[c]Aged between 13 and 24.
[d]Aged between 25 and 64.
[e]Aged 65 or older.
[f]Measured by capita expenses proportional to the baseline minimum wage (MW). Level 1 (More wealth): "1 or more". Level 2: "0.5 to 1". Level 3: "0.25 to 0.49". Level 4: "0<to 0.24". Level 5 (Lower wealth): "Nothing declared."
[g]Water supply: Other – well, spring, and others.
[h]Housing Material: No – Coated clay, uncoated clay, wood, and others.
[i]Lighting: Non-electric – No meter, lamps, candles, and others.
[j]Average rates for the period (2007-2015) by municipality.
[k]% of the municipal population with inadequate baseline sanitation.
[l]Baseline municipal unemployment rate.
[m]Per 1,000 inhabitants of the baseline municipality. All statistical tests used where two-sided and, where appropriate, adjustments were made for multiple comparisons.

**Table 3 | Estimates of the average effect of the *Programa Bolsa Família* (PBF), in adjusted IPTW Poisson models (with robust standard errors), on AIDS incidence, mortality, and the case-fatality rate, 2007-2015, by wealth, sex, and age subpopulation**

| Adjusted Models | Incidence | | Mortality | | Case-Fatality | |
|---|---|---|---|---|---|---|
| | RR[a] | 95% CI[b] | RR[a] | 95% CI[b] | RR[a] | 95% CI[b] |
| Wealth[c] | | | | | | |
| Quartile 1[d] (Lower wealth) | 0.45*** | (0.42-0.47) | 0.46*** | (0.42-0.49) | 0.63*** | (0.51-0.76) |
| Obs. | 5,503,505 | | 5,503,510 | | 4,059 | |
| Quartile 2[e] | 0.63*** | (0.59-0.67) | 0.65*** | (0.58-0.73) | 0.77* | (0.60-0.98) |
| Obs. | 3,868,703 | | 3,868,708 | | 2,398 | |
| Quartile 3[f] | 0.84*** | (0.77-0.91) | 0.84** | (0.72-0.96) | 1.05 | (0.80-1.38) |
| Obs. | 5,051,629 | | 5,051,636 | | 1,880 | |
| Quartile 4[g] (More wealth) | 1.00 | (0.86-1.15) | 0.98 | (0.77-1.25) | 0.93 | (0.60-1.42) |
| Obs. | 5,158,593 | | 5,158,596 | | 1,629 | |
| Sex | | | | | | |
| Female | 0.60*** | (0.57-0.63) | 0.58*** | (0.53-0.62) | 0.65*** | (0.53-0.78) |
| Obs. | 11,309,111 | | 11,309,119 | | 5,289 | |
| Male | 0.61*** | (0.57-0.63) | 0.64*** | (0.59-0.70) | 0.88 | (0.74-1.03) |
| Obs. | 8,268,518 | | 8,268,530 | | 4,676 | |
| Age | | | | | | |
| Adolescents and young people[h] | 0.48*** | (0.44-0.51) | 0.46*** | (0.39-0.54) | 0.72 | (0.46-1.12) |
| Obs. | 5,258,575 | | 5,258,576 | | 1,805 | |
| Adults[i] | 0.62*** | (0.60-0.64) | 0.62*** | (0.57-0.65) | 0.75*** | (0.65-0.85) |
| Obs. | 12,405,560 | | 12,405,579 | | 7,973 | |
| Older people[j] | 1.18 | (0.78-1.78) | 1.54 | (0.91-2.59) | 0.06 | (0.00-4.10) |
| Obs. | 1,913,494 | | 1,913,494 | | 154 | |

***p value < 0.001; **p value < 0.01; *p value < 0.05.
[a]Incidence-rate ratios.
[b]Confidence interval.
[c]Measured by capita expenses proportional to the baseline minimum wage (MW).
[d]Quartile 1: 0% to 0.1%.
[e]Quartile 2: 0.1% to 18.5%.
[f]Quartile 3: 18.5% to 56.5%.
[g]Quartile 4: more than 56.5%.
[h]Aged between 13 and 24.
[i]Aged between 25 and 64.
[j]Aged 65 or older. All models were adjusted for the same demographic and socioeconomic variables in Table 2. All statistical tests used where two-sided and, where appropriate, adjustments were made for multiple comparisons.

## Complementary, sensitivity and triangulation analyses

In the appendix (pages 38–43), we present complementary analyses that show the robustness and plausibility of our results. One significant finding is illustrated in Table S24, which challenges the perception that the Programa Bolsa Família (PBF) exclusively targets formal families. According to PBF eligibility rules[25] and reports[26] the beneficiaries of the program encompass a diverse range of individuals, including those in low-income or extremely low-income, whether single or in couples, even if they do not have children. Within our study cohort, only 22.7% of PBF beneficiaries are married, similar to the percentage found among the non-beneficiaries (28.7%). Additionally, the proportion of single adults is slightly higher among PBF beneficiaries (71.7%) compared to non-beneficiaries (63.7%). These percentages can be attributed to the breakdown of traditional family structures within the most impoverished communities[27]. We conducted a complementary analysis where we included marital status – not considered in the main models due to the high proportion of missing values – as an adjusting variable, showing no statistically significant differences with the main models (Table S25).

As further complementary analysis, we evaluated the variable of sexual preference, available only in the dataset of People Living with AIDS (PLWA): the proportion of Men who have Sex with Men (MSM) within the PLWA cohort was accounting for 15.0% of the 26,936 PLWA individuals, suggesting that HIV transmission within this cohort of economically disadvantaged Brazilians could be in a large part due to heterosexual relationships. Furthermore, when comparing the percentage of MSM between PBF beneficiaries and non-beneficiaries within the PLWA cohort, the numbers were almost identical (Table S26). Upon introducing the variable of AIDS exposure category, which includes MSM, in the assessment of PBF effects on AIDS case-fatality rates (the only analysis conducted within the PLWA cohort), the results were once again similar to the previous models, with no statistically significant differences observed (Table S27). Although we were unable to adjust the models of AIDS incidence and mortality for the variable of sexual behavior (which was only available within the PLWA cohort), we can assume that it would not significantly alter the results based on the estimations mentioned earlier and the small percentage of MSM within the study cohort. While further stratification analyses of extremely vulnerable populations, such as incarcerated individuals and users of illicit drugs, have not been possible due to the lack of these variables in the study cohort, it is plausible that PBF could exert a particularly strong effect on these populations, at least of similar magnitude what have been shown in extremely low-income individuals.

The main factors attributed to the effects of PBF include increased access to the healthcare system, HIV diagnosis, prevention of AIDS progression, and treatment of AIDS (which also reduces HIV transmission in the community), as well as improved compliance to

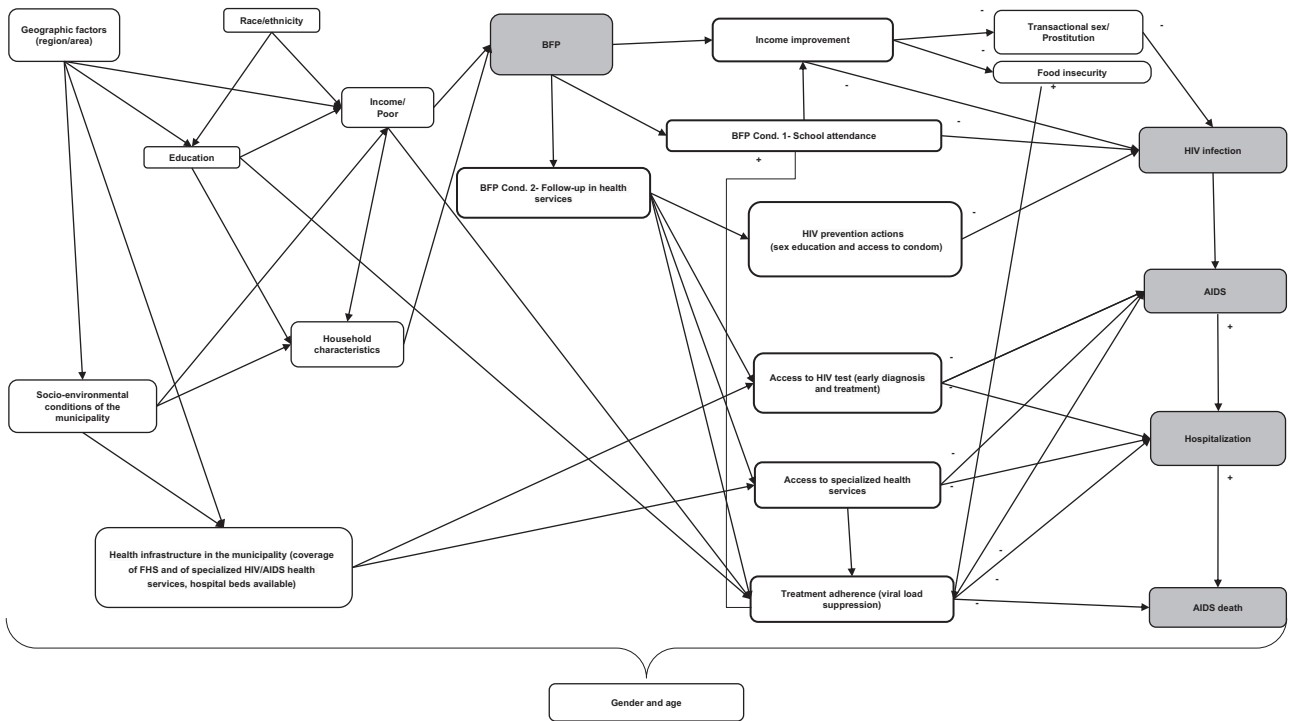

**Fig. 2 |** Conceptual model of the potential pathways through which *Programa Bolsa Familia* (PBF) can affect AIDS incidence, mortality, and case-fatality rates.

treatment due to reduced economic and geographic barriers, and improved nutritional and psychological status. In Brazil, until 2013, only AIDS cases had mandatory notification. Thus, we chose to exclude HIV notifications, which before 2013 were extremely underreported – and the main analyses were referred only to AIDS cases. However, despite these limitations and as complementary analysis, we estimated the association between the PBF exposure and the HIV cases notified in the database after 2013, and we found that receiving PBF benefits was associated to a 36% decrease in the HIV notification rate (RR: 0.64; 95% CI: 0.62–0.66) (Table S28).

As shown in Fig. 2, the gradient of effects according to the levels of wealth is extremely consistent across all HIV/AIDS indicators, as expected considering that they are sequential indicators of the progression of the disease in which poverty, and poverty-relief interventions, could act in similar ways. In Fig. 3 we show the visual comparison of the effect estimates – obtained through the adjusted IPTW Poisson regressions described above - for the four HIV/AIDS related outcomes, that is HIV notification (only starting from 2013), AIDS incidence, AIDS mortality, AIDS case-fatality (all starting from 2007), in the overall study population and in the quartiles of wealth. Moreover, the richest quartile (4th) shows no statistically significant effects on all indicators, demonstrating not only the robustness of the gradient but also acting as negative control for each outcome under study[24]. As a matter of fact, if the observed statistically significant associations of the PBF with HIV and AIDS-related outcomes were due to omitted variable biases, or other kind of biases related to the study design, they should have persisted in the analyses among individuals with higher income.

To address any potential limitations in our study design, it is crucial to clarify that the assumption of the PBF rules selecting the more "motivated" individuals and leaving the vulnerable population in the "non-benefit" comparator group does not apply. This is because, as described in the methods section, the participants of this study are only individuals registered in the *Cadastro Único*, and after this registration process, the centralized system of the Ministry of Social

Development enrolls them into the PBF based on specific eligibility rules (see methods section) with no influence during this enrollment phase of any condition or CCT conditionality[25]. Furthermore, according to our study design, all enrolled PBF beneficiaries who have received allowances are considered exposed to the PBF, even if they leave the program before the end of the study period (which occurred in 27.7% of the cases). This approach allows us to account for the post-exposure effects of PBF, but also avoid that "less motivated" individuals - who leave the program due to incompliance with PBF conditionalities – could bias our effect estimates. As a matter of fact, while a majority of individuals leave the PBF due to an increase in their income above the PBF eligibility thresholds[26,28], only a small portion leaves due to non-compliance with conditionalities[29]. Nevertheless, these individuals are still considered part of the exposure group in our analysis.

To verify the robustness of the results, we systematically performed a wide range of sensitivity analyses (see the methods section and appendix p.21-33) for each outcome. First, we included different municipal-level variables, second, we tested the influence of IPTW, third, we evaluated the relevance of the endemic levels of AIDS, fourth, we tested different proxies of wealth, fifth, we fitted the same models with different specifications, including different individual-level covariates and robust standard errors, sixth, we evaluated models according to the quality of information, and finally, we evaluated the influence of missing values. Moreover, in order to verify the degree of confidence in the causal inference and in the findings of the impact evaluation, we did triangulation analyses (see appendix p. p.34-36) using alternative methodologies[30], including survival models and propensity score matching (PSM), and all triangulation analyses indicated an high degree of confidence in the results[30].

## Discussion
To the best of our knowledge, this is the largest and most comprehensive impact evaluation of the effects of a Conditional Cash Transfer program on an infectious disease, and in particular on HIV/AIDS. We

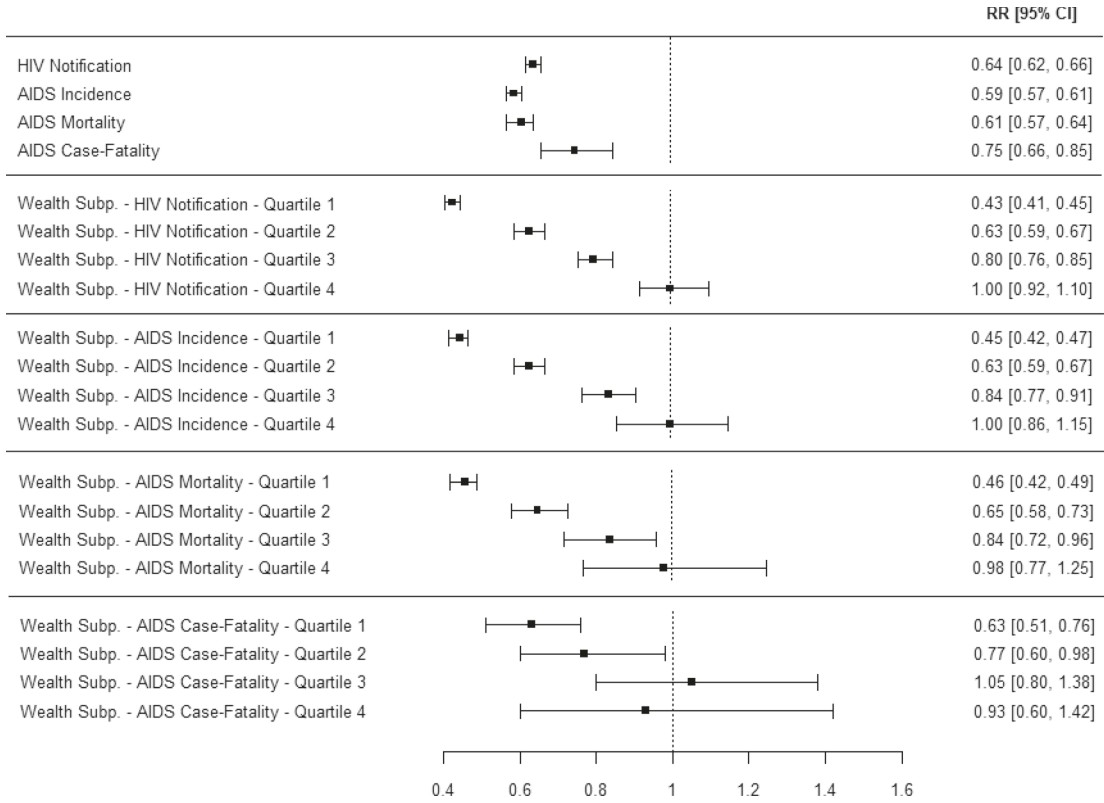

The samples size for each adjusted IPTW Poisson models: HIV Notification ($n = 5,983$); HIV Notification - Quartile 1 ($n = 1,959$); HIV Notification - Quartile 2 ($n = 1,263$); HIV Notification - Quartile 3 ($n = 1,398$); HIV Notification - Quartile 4 ($n = 1,349$). AIDS Incidence ($n = 19,577,629$); AIDS Incidence - Quartile 1 ($n = 5,503,505$); AIDS Incidence - Quartile 2 ($n = 3,868,703$); AIDS Incidence - Quartile 3 ($n = 5,051,629$); AIDS Incidence - Quartile 4 ($n = 5,158,593$). AIDS Mortality ($n = 19,577,649$); AIDS Incidence - Quartile 1 ($n = 5,503,510$); AIDS Incidence - Quartile 2 ($n = 3,868,708$); AIDS Incidence - Quartile 3 ($n = 5,051,636$); AIDS Incidence - Quartile 4 ($n = 5,158,596$). AIDS Case-Fatality ($n = 9,965$); AIDS Incidence - Quartile 1 ($n = 4,059$); AIDS Incidence - Quartile 2 ($n = 2,398$); AIDS Incidence - Quartile 3 ($n = 1,880$); AIDS Incidence - Quartile 4 ($n = 1,629$).

**Fig. 3 | The samples size for each adjusted IPTW Poisson models.** Comparison of the average effect of the *Programa Bolsa Família* (PBF), obtained through adjusted IPTW Poisson models with robust standard errors, on HIV notification, AIDS incidence, AIDS mortality, and AIDS case-fatality rate, 2007–2015, in the overall study population (first four lines) and by quartiles of wealth for each outcome.

were able to show the significant impact of the world's largest CCT on all sequential AIDS-related outcomes – namely incidence, mortality and case-fatality rates - among low-income individuals in a LMIC. Moreover, similar effects were found on HIV notification. Combining big data from a nationwide cohort of 22,788,998 individuals, including 22,212 AIDS cases and 7,950 AIDS deaths over a 9-year period, with a robust quasi-experimental impact evaluation design and a wide range of sensitivity and triangulation analyses, we found a strong effect of PBF on the reduction of AIDS incidence (41%), mortality (39%) and the case-fatality rates (25%). Interestingly, the impact of the CCT was concentrated in the extremely low-income individuals of the cohort, showing a gradient of effectiveness based on the levels of socio-economic vulnerability of the beneficiaries. Moreover, in the highest income quartile, PBF had no statistically significant effects in any of the outcomes. The impact of PBF was also stronger among women and adolescents, demonstrating the potential of CCTs in reducing health inequalities based not only on socioeconomic position, but also on sex and age.

CCT money allowances aim to promptly alleviate socioeconomic vulnerabilities, while conditionalities ensure that these families will be able to break the vicious cycle of poverty in the long term[8,16]. As in the large majority of CCTs, PBF conditionalities are focused on the vulnerable individuals of the family: for the health-related conditionalities pregnant women should attend all prenatal consultations, and mothers all educational activities for breastfeeding and child health, while children need to comply with the regular vaccination schedule and routine check-ups for growth and development[8,16]. Educational conditionalities are also focused on children, adolescents and potential young mothers, which should have a school attendance above established thresholds[8,16].

Several studies have already shown that CCTs increase the use of preventive services, promote healthy behavior, and improve a wide range of health outcomes[8]. In fact, CCTs were also able to reduce the incidence of other infectious diseases, such as tuberculosis[31,32] and leprosy[33], and decrease maternal and child mortality[16]. However, regarding HIV/AIDS, previous studies on the impact of cash transfers have shown inconsistent results:[4] while a randomized controlled trial in Malawi showed an HIV prevalence odd ratio of 0.36 between beneficiaries and non-beneficiaries of a cash transfer intervention for young women[10], another randomized trial in South Africa demonstrated no statistically significant effects on HIV notification[34]. Small conditional economic rewards have also been associated to a reduction in sexually transmitted diseases in Tanzania[35]. A recent systematic review of the effects of cash transfer programs on HIV prevention suggested that cash transfers for vulnerable families may reduce risky behavior in adolescents (especially girls), but pointed out the mixed

and inconsistent results from the literature, arguing for large-scale impact evaluations able to include the most vulnerable populations[4]. Regarding AIDS-related outcomes, the evidence is scarce: a randomized trial showed that incentives can promote short-term adherence at the time of HIV treatment initiation[36], while a recent ecological observational study showed a relevant association between CCT coverage and reductions on AIDS-related hospitalizations and deaths[37].

Our results, obtained using a number of individuals, cases, and deaths from AIDS by far larger than any other previous study, provide a possible explanation for such inconsistencies, showing that CCT impact largely depends on the baseline income levels of the beneficiaries, and it is considerably stronger for extremely low-income people in comparison with higher income individuals, among which CCT effects can even be insignificant. Moreover, our results show that CCT effects are more pronounced among women and adolescents, which have to comply with the program's conditionalities.

CCTs have the potential to play a significant role in preventing AIDS through various mechanisms (Fig. 2)[4,8,10,34,37]. These mechanisms include the positive impact of increased household income and the conditionalities associated with education and healthcare that are necessary to receive the benefits. The reduction of AIDS-related morbidity and mortality can be attributed to several factors. Firstly, CCTs improve the socioeconomic conditions of families, particularly in situations of extremely low-income. By providing cash transfers, these programs can decrease the likelihood of women resorting to sex work or engaging in transactional sex for basic needs, thereby empowering them economically and enhancing their self-esteem[4,10,11]. Additionally, CCTs contribute to the reduction of risky sexual behavior, thereby preventing new cases of HIV infection and transmission[38]. The monitoring of women and children by healthcare teams and social assistance it also aids in the management of comorbidities in the late stages of AIDS, provides access to essential health information regarding treatment adherence, reduces vertical transmission, and prevents transmission to other partners[4].

Secondly, implementing education conditionalities can have a positive impact on school enrollment and attendance, particularly for girls, thereby serving as protective measures against HIV infections[4,11]. Thirdly, by incorporating health conditions, such as regular prenatal and pediatric visits, these programs can encourage females to seek healthcare services and engage in sexual education activities[4,37]. The implementation of CCTs can also play a crucial role in slowing down or preventing the progression from HIV infection to AIDS, as well as AIDS-related deaths. By facilitating early detection and monitoring through specialized HIV/AIDS services, CCTs can address economic and geographical obstacles, enhance adherence to antiretroviral treatment, and achieve viral suppression, thereby reducing AIDS incidence, mortality rates, and HIV transmission in the general population[4,11,37]. Moreover, by granting individuals with HIV increased access to healthcare services and ensuring cost-free access to comprehensive care and ART, CCTs promote self-care and enhance adherence to treatment[37]. Furthermore, these programs alleviate food insecurity and malnutrition, both of which compromise the immune system and can hinder the transition from HIV to AIDS, as well as the effectiveness of antiretroviral treatment[37,39]. Our findings reinforce these ideas, as the most significant impact observed was a reduction in AIDS-related incidence and mortality rates, followed by a decrease in case-fatality rate.

Our study has limitations. Firstly, although we estimated the effect of PBF by controlling for a large number of confounding variables and using IPTW, propensity score-based models are not able to control for unobservable confounding variables. For this reason, for each outcome, we included the average municipal rate of the AIDS indicator over the period (incidence, mortality, and fatality) as an independent variable in the logistic and in the Poisson regressions, allowing adjustment for the baseline levels of the specific AIDS outcome,

together with potentially associated unobservable variables. Moreover, the extended sensitivity analyses showed that the inclusion of other independent variables related to socioeconomic inequalities, healthcare services, and municipal infrastructure, did not affect the results of the PBF effect estimates (appendix, pp.22-24). Second, due to the limitations in the data sources, it was not possible to adjust for the risky sexual behaviors of the individuals. However, these risk factors should be considered mediators -and not confounders- of the PBF effects and of the social determinants included in the models, consequently, their inclusion in the regressions would lead to an over-adjustment of the models[32,37].

Third, our cohort – obtained from the linkage of the Unified Registry for Social Programs with AIDS-related health records - contains data only from individuals recorded in the Unified Registry, which are part of the half of the Brazilian population with lower incomes. However, while high-income individuals are absent in our cohort, lower middle-income individuals, which need to be recorded in the Unified Registry to get access to different governmental assistance programs, are included - as shown in our stratified analyses – even if underrepresented. As a consequence, while our stratified analyses allowed to evaluate PBF effects on a wide range of different subpopulations, our overall estimates are representative of the lower income half of the country.

The main strength of our study is the combination of an extremely large longitudinal dataset with robust quasi-experimental impact evaluation methods, offering a unique opportunity to evaluate the effects of interventions with an unprecedented number of individuals and subpopulations, usually not included or underrepresented in traditional epidemiological studies and randomized controlled trials. This is particularly relevant in policy evaluation: as shown in our study, public interventions could have a very different impact according to the characteristics and baseline conditions of their beneficiaries. Another important strength of our study is the wide range of sensitivity analyses performed, which confirmed the robustness of the findings, and the use of different triangulation analyses, that demonstrate a high degree of confidence in the results of the impact evaluation.

Our study, taking advantage of a large cohort of the low-income individuals in a LMIC, was able to show how conditional cash transfer programs could significantly reduce morbidity and mortality from AIDS among socioeconomically vulnerable populations, and how their impact depends on income levels. In the current context of global increase of poverty due to consequences of the COVID-19 pandemic, the war in Ukraine, and the global inflationary surge, a strengthening and expansion of CCTs in LMICs could significantly reduce AIDS morbidity and mortality among the most vulnerable populations, and substantially contribute to the achievement of the AIDS-related Sustainable Development Goal.

## Methods

### Study design, population, and ethical issues

This study has a quasi-experimental cohort study design, based on the longitudinal information of 22.7 million individuals aged 13 and older, from January 1 2007 to December 31, 2015. This study was approved by the Research Ethics Committee of the Institute of Collective Health of the Federal University of Bahia (ISC/UFBA), under number 41691315.0.0000.5030 (Report No:3.783.920).

### Data sources, outcomes, and intervention

The 100 Million Brazilian Cohort was based on the baseline information of families, during the period from January 1, 2001 to December 31, 2017, who sought to benefit from the Brazilian government's social programs through registration in the Unified Registry for Social Programs (in Portuguese: *Cadastro Único para Programas Sociais – Cadastro Único*). The Unified Registry is an administrative database, to which Brazilians aged 16 or over can apply by registering their personal

information (age, sex, race/ethnicity, education and others) and household information (household density, familiar income, structural characteristics of the residence and others), as long as they are within one of these categories: (i) belong to a family with a monthly per capita income of up to half a minimum wage; (ii) belong to a family with a total monthly income of up to three minimum wages; (iii) belong to a family with an income greater than three minimum wages, provided that the registration is linked to inclusion in social programs in the three spheres of government; (iv) be the only resident of the household, or; (v) living on the streets (alone or with the family).

Upon registration, individuals receive a unique identifier code and are searched for socioeconomic characteristics. At the end of 2017, Unified Registry had approximately 114 million individuals on its register, which represents around 50% of the Brazilian population. It is a social tool that identifies and characterizes especially low-income families, allowing the government to know the socioeconomic aspect of the poorest and use it for the selection of social programs[20,40].

Two individual-level health-related datasets were linked to the Unified Registry for Social Programs: the Notifiable Diseases Information System (SINAN) and the Mortality Information System (SIM)[22]. SINAN is a decentralized information system that monitors the incidence of notifiable diseases, including HIV/AIDS[22]. SIM is a national death surveillance system that registers deaths by all causes, including HIV/AIDS, according to CID-10 classification.

Created by the Center for Integration of Data and Knowledge for Health (CIDACS / FIOCRUZ)[20], the Cohort aims to facilitate research and continuous assessment of social determinants and the effects of social policies and programs in health contexts in Brazil. It has 246 variables with demographic and socioeconomic information at the individual and family level. The codes and linking algorithms between the databases were built to make efficient and specific links through five identifiers: the date of birth, the municipality of residence, the sex, the name and the mother's name of each individual presented in each of the databases[20,23,41]. The linking codes and algorithms were built based on five identifiers: date of birth, place of residence, sex, name and mother's name of the individual present on each database[20,23]. The Unified Registry and the health datasets (SIM and SINAN) were individually matched in two steps, using the CIDACS-RL tool (appendix,p.3)[20,23]. The quality of each link between Unified Registry, SINAN e SIM has been extensively evaluated and validated[16,19].

The 100 Million Brazilian Cohort resulting from this linkage has undergone several verification and consolidation processes[16], and it has been used in various impact evaluation studies related to social determinants of health and PBF[13,20]. An aggregated dataset - containing municipal-level information on AIDS endemicity levels, municipal infrastructures, and healthcare resources - was also deterministically linked to the Unified Registry through the individuals' municipal code of residence and baseline year.

The AIDS outcomes available in the final dataset were: (i) new AIDS cases, defined by adapted CDC criteria, the Rio de Janeiro/Caracas criteria;[15] and (ii) AIDS deaths, considering as underlying cause the CID-10 codes B20 to B24[15].

The follow-up time for each individual in the cohort, i.e., person-years, started on the date of entry into the cohort until the date of AIDS diagnosis (for AIDS incidence rate), the date of death due to AIDS (for AIDS mortality rate), or, for individuals without any AIDS-related outcome, the date of death from other causes or the end date of the cohort (December 31 2015). For AIDS case-fatality rate, the start date began with the date of diagnosis and ended with the AIDS-related death.

The beneficiary group was defined as eligible individuals who received PBF benefits [extremely low-income families are considered to have a monthly per capita income of up to US$12 (BRL 60) in 2007-2008 period, up to US$14 (BRL 70) from 2009 to 2013, and up to US$15.4 (BRL 77) in 2014 and 2015. Low-income families are those with a per capita income between US$12 (BRL 60.01) and US$24 (BRL 120) in 2007 and 2008, between US$14 (BRL 70.01) and US$28 (BRL 140) from 2009 to 2013, and between US$15.4 (BRL 77.01) and US$30.8 (BRL 154) in 2014 and 2015], and their exposure started with the receipt of the benefit, until the end of their cohort follow-up. The non-beneficiary group was defined as individuals who had never benefited from PBF throughout their follow-up period, as in previous studies[13]. In case of administrative delays and non-receipt of the benefits, eligible individuals were classified in the non-beneficiary group (details provided in the appendix p.4). Administrative delays occur because after registering with *Cadastro Único* and fulfilling the aforementioned requirements, the Ministry of Citizenship (in Portuguese: *Ministério da Cidadania*) elects the beneficiaries of the program.

## Statistical analyses

To estimate the association between PBF exposure and AIDS incidence, mortality, and case-fatality rates we used multivariable Poisson regression models, adjusted for all relevant demographic, socioeconomic, and healthcare-related confounding variables at the individual and municipal level, with follow-up time as an offset variable, robust standard errors, and observations weighted through stabilized, truncated, Inverse Probability of Treatment Weighting (IPTW). Poisson regression models with IPTW are widely used in quasi-experimental cohort studies which investigate the impacts of public and social policies on health outcomes[42–44], including evaluation studies that used the 100 Million Brazilian Cohort dataset[32,33].

The process of IPTW consists of two primary stages. Initially, the likelihood, also known as the propensity, of encountering the risk factor or intervention under consideration is computed based on an individual's attributes (i.e., propensity score). Subsequently, weights are calculated as the inverse of the propensity score. By applying these weights to the study's population, a simulated population is generated where potential sources of bias are equally balanced between the exposed and unexposed groups[45].

We consider a scenario with two potential treatments: one involving exposure to PBF (treatment) and the other being unexposed (control). Within the potential outcomes framework, each individual is associated with a pair of potential outcomes (AIDS outcomes): $Y_i(1)$ and $Y_i(0)$. These outcomes represent the results under the treatment and control conditions, respectively, when subjected to the same conditions[44]. However, each individual is assigned either the treatment or control, but not both. We denote $\mathbf{Z}$ as an indicator variable for treatment (exposure to PBF) ($Z = 1$ for treatment and $Z = 0$ for control). Consequently, only a single outcome, $Y_i$, is observable for each individual: this is the outcome linked to the actual treatment they received. The observed outcome, denoted as $Y_i$, is determined by $Y_i = Z_i Y_i(1) + (1 - Z_i) Y_i(0)$. Thus, $Y_i$ is equivalent to $Y_i(0)$ if $Z_i(0)$, and is equivalent to $Y_i(1)$ if $Z_i(1)$[44]. Let $\mathbf{X}$ represent a vector of observed baseline covariates.

Initially, we used logistic regression to get the propensity score. The logistic regression equation models the probability of a binary outcome (usually 0 or 1) as a function of one or more predictor variables. The logistic function, also known as the sigmoid function, is used to transform the linear combination of predictors into the probability of the event occurring[46,47].

$$P(Z = 1 | X) = \frac{1}{1 + e^{-(\beta_0 + \beta_1 X_1 + \beta_2 X_2 + \cdots + \beta_p X_p)}} \qquad (1)$$

where $P(Z = 1 | \mathbf{X})$ is the probability of the binary outcome being 1 given the predictor variables $\mathbf{X}$; $e$ is the base of the natural logarithm; $\beta_0, \beta_1, \beta_2, \ldots, \beta_p$ are the coefficients corresponding to the predictor variables $\mathbf{X_1, X_2, \cdots X_p}$. We estimated the probability of each individual to receive PBF (propensity score-PS), in two ways[43,44,48]. For the first equation, we calculated the marginal probability of treatment ($PS_t$)

and then we estimated the multivariable PS ($PS_{mul}$), adjusted for all relevant covariates.

We used $PS_t$ and $PS_{mul}$ as weights to calculate the stabilized IPTW using the formulas:

$$w_{Z=1} = \frac{PS_t}{PS_{mul}} \tag{2}$$

$$w_{Z=0} = \frac{(1 - PS_t)}{(1 - PS_{mul})} \tag{3}$$

where $w_{Z=1}$ is the weights for the beneficiaries, and $w_{Z=0}$ is the weights for the non-beneficiaries. In order to correct for possible extreme weights we set thresholds, with weights exceeding the set value converted to that threshold value. In this investigation, the weights were truncated based on the distribution of their values for the 1st and 99th percentiles, which represented these thresholds[43,44,48]. IPTW uses the propensity score to balance baseline characteristics in the exposed and unexposed groups by weighting each individual by the inverse probability of receiving treatment[45].

The Poisson equation with IPTW is a framework used to analyze count data or event occurrences, while accounting for the potential bias introduced by non-random treatment assignment in observational studies. In this context, the Poisson equation models the relationship between the event outcomes and covariates while incorporating IPTW to adjust for treatment selection bias.

$$\log(Y_i) = \beta_0 + \beta_1 X_{i1} + \beta_2 X_{i2} + \cdots + \beta_p X_{ip} + \log(IPTW_i) \tag{4}$$

where $\log(Y_i)$ is the natural logarithm of the rate for individual $i$; $X_{i1}, X_{i2}, \ldots, X_{ip}$ are the covariates for individual $i$; $\beta_0, \beta_1, \ldots, \beta_p$ are the coefficients associated with the covariates; and $\log(IPTW_i)$ represents the logarithm of the inverse probability treatment weight for individual $i$. Finally, multivariable Poisson regressions, adjusted by stabilized, truncated IPTW, were estimated with the same socioeconomic and demographic variables adopted in the logistic model for all AIDS outcomes.

At the individual level, the demographic and socioeconomic and variables included in the models were sex, race/ethnicity, age, educational achievement, per capita expenditures used as a proxy for wealth, household characteristics (adequate water supply, household construction material, and installed electric power), geographic region, area of residence (urban and rural), and year of entry into the cohort. Due to the ART potential mediator effects in the relationship between PBF coverage and mortality from AIDS, we have also fitted the case-fatality rate models without ART as independent variable, with no changes in the PBF estimates (see appendix, Table S30, page 45). As municipal-level variables, we included the proportion of the population with inadequate sanitation, the unemployment rate, and a set of healthcare service-related variables (calculated as a rate per 1,000 inhabitants in the municipality): the number of doctors, nurses, and hospital beds. Lastly, in order to control for any selection bias and unobserved confounding associated with municipal endemic levels of AIDS, the average municipal AIDS incidence rate over the study period was included as a covariate in the AIDS incidence models. For mortality and case-fatality rates, we also included the municipal average AIDS mortality and case-fatality rates, respectively.

In order to investigate heterogeneity in the impact of PBF participation, the same estimation process was performed for different population strata: per capita wealth (stratified by quartile of the sum of per capita expenditure), sex, and age (adolescents and young people versus adults and the older people).

In order to assess the robustness of the results, we performed a wide range of sensitivity analyses (appendix, pp.21-36). Firstly, to evaluate the relevance of variables that capture municipal-level characteristics, we fitted models only with individual-level variables, and tested the inclusion of a wide range of different aggregate-level variables. Secondly, to understand the influence of IPTW on the PBF effect estimations, we estimated and compared all models without IPTW. Thirdly, to evaluate the influence of the inclusion of the endemic levels of AIDS in the models, we fitted the same regressions without the average municipal rates for each outcome (incidence, mortality, and case-fatality rate). Fourthly, to evaluate the adoption of per capita expenditure as a proxy of wealth, we ran the same analyses with other proxies, such as per capita income. Fifth, to evaluate the robustness of the results, we fitted the same models with different specifications, including different sets of individual-level covariates, variable distributions, and without robust standard errors, among others. Sixth, to evaluate the potential influence of low-quality information on our findings, we estimated and compared the models selecting only individuals living in municipalities with vital information of adequate quality, according to consolidated criteria[16,49]. Seventh, to evaluate the influence of missing values on our findings, we estimated the association of PBF for all individuals, including participants with missing data for one or more variables, both for estimating PS and adjusted Poisson models. Finally, to have a high degree of confidence in the causal inference and in the findings of the impact evaluation, we did triangulation analyses using alternative methodologies[50], including survival models and propensity score matching (PSM).

### Reporting summary

Further information on research design is available in the Nature Portfolio Reporting Summary linked to this article.

### Data availability

The protocol for the creation of the 100 Million Brazilians Cohort and the cohort profile of the 100 Million Brazilians Cohort is available in the publications referenced in the article and further material is available at: https://cidacs.bahia.fiocruz.br/en/platform/cohort-of-100-millionbrazilians. The linkage protocols are explained in the referenced publications and the codes are available at: https://gitHub.com/gcgbarbosa/cidacs-rl. However, the datasets generated during and analyzed during the current study are not publicly available due to confidentiality and ethical issues. To request access, contact us at: https://cidacs.bahia.fiocruz.br/contato/fale-conosco/.

### Code availability

The code can only be shared on request due to confidentiality and ethical issues. To request access, contact us at: https://cidacs.bahia.fiocruz.br/contato/fale-conosco/.

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

## Acknowledgements

This study was supported by the US National Institute of Allergy and Infectious Diseases, National Institutes of Health (grant number 1R01AI152938). We thank the data production team and all Center for Data Integration and Knowledge in Health—FIOCRUZ collaborators for their work on building the 100 Million Brazilians Cohort. We thank our colleagues from the Collective Health Institute (Federal University of Bahia, Salvador, Brazil) for their valuable contributions during the development of the study. We acknowledge support from the grant CEX2018-000806-S funded by MCIN/AEI/ 10.13039/501100011033, and support from the Generalitat de Catalunya through the CERCA Program

## Author contributions

D.R., I.D. and J.M. developed the study concept. M.Y.I., M.L.B. and A.F.S. collected the data. D.R., A.F.S., I.L., J.P., G.S.J. and D.B.M. designed the study and investigation. A.F.S., D.R., I.L. and N.S.G. did the data analysis and wrote the first draft of the manuscript. C.A.S.T.S., L.M., L.E.S., G.A.S.M. and R.V.R.A. have contributed to the first draft of the manuscript. All authors contributed to data interpretation and reviewed and edited the manuscript. D.R., I.S. and J.M. supervised the study process.

## Competing interests

The authors declare no competing interests.
