## [Peer Review File · Nature Communications]

Impacts of conditional cash transfers on HIV/AIDS incidence, mortality, and case fatality rates in BrazilREVIEWER COMMENTS

Reviewer #1 (Remarks to the Author):

Overall this is an impressive analysis of a very large cohort and the results of the PBF on AIDS outcomes are large. Very important findings. That said, there are some issues with the cohort construction and variable description that if better clarified could strengthen the paper. Overall being more clear about eligibility for the PBF in the paper (not just linking to a citation), being more clear in main text about entry/eligibility into the primary data set (from which it seems health metrics were linked) and being more clear about metrics (e.g. HIV incidence?) - these will significantly strengthen the paper.

Please better describe who the comparison group is in this analysis- those who didn't get PBF but if the primary data set is everyone in the social protection data set- why would they not have been eligible? Just be more clear about study design and comparator in the main text.

While the authors cite original publications for the 100 million Brazilian cohort- how was that actually constructed?. It would help the reader to briefly say something on how individuals entered into that cohort (how were they identified, eligibility) Is everyone in it eligible for the PBF? This was not totally clear given the statement that it was created from the unified registry for social protection (line 136). Also just stating in the text the eligibility criteria for the PBF would also help in understanding who is and is not in the data set- the citation included (Rasella Lancet 2013) seems to suggest the program is for families that have children or pregnant women so if this is the case it would suggest the target population for this analysis is more focused than suggested. Please state who is eligible.

Please clarify if HIV incidence was measured and how? I see new AIDS diagnoses which I understand it being called AIDS incidence- would prefer it just be called new AIDS diagnoses.

To the point about HIV incidence "HIV cases notified 235 in the database after 2013, and we found that receiving PBF benefits was associated to a 236 36% decrease in the HIV incidence rate (RR: 0.64; 95% CI: 0.62-0.66) (Table S28)." Its not clear this is HIV incidence, again it is new HIV diagnoses (yes...new? Not sure HIV diaganoses) but it doesn't mean the infection is a new infection does it? It could be the person has been infected for years but was just diagnosed. Unless using a test that captures incidence? Seems not. Again use language that more accurately represents measure (HIV tests?)

Line 215 "proportion of Men who have Sex with Men (MSM) within the PLWA cohort was relatively 215 low, accounting for only 15.0% of the 26,936 PLWA individuals, indicating that HIV transmission within this cohort of economically disadvantaged Brazilians was primarily due 217 heterosexual relationships." I would be careful about this statement- again it depends on the sampling frame and who was entered into the original data set for linkage- it if individuals eligible for PBF is there some bias in who is in the dataset (seem likely since many new HIV infections in Brazil are among MSM)- raising questions again as to whether social protections are reaching key populations- I am not sure this data set can answer this question since there is selection bias (yes?) in who is in the original data?

Reviewer #2 (Remarks to the Author):

Authors report an analysis assessing the causal effects of a large conditional cash transfer program in Brazil (Programa Bolsa Familia - PBF) on several important HIV-related outcomes, finding reductions in AIDS incidence, AIDS-related mortality, and AIDS case-fatality. They also find suggestive evidence that PBF leads to reductions in HIV incidence. Causal claims are further supported by a gradient effect by wealth quintile, whereby the strongest effects are seen among

the poorest participants. This is a very important, innovative, and unique study that is able to connect several very large datasets and assess the effects of an exemplary conditional cash transfer program on key HIV outcomes. I generally found the methodology to be of the highest standard, with extensive sensitivity analyses addressing many of my initial concerns. This work adds to the evidence that social protection and poverty reduction programs are going to be critical components of efforts to end HIV epidemics now that we are in an era of widespread access to antiretroviral therapy. I have a few suggestions for the authors to consider that I believe would strengthen this work.

1. Some references are a bit outdated in the introduction. Consider putting in the context of more recent systematic reviews of cash transfer effects on general outcomes (Bestagli et al, ODI 2016) and HIV outcomes (Guimarães et al, Lancet HIV 2023 — I believe co-authored by some of the authors of this manuscript), and our recent paper directly evaluating cash transfer effects on HIV-related outcomes in 42 LMICs (Richterman and Thirumurthy, Nature Human Behaviour 2022).

2. Line 102 — I imagine the income limits and benefits listed have not been static over the 20 year program history, are these figures from a specific year? Please specify what year they come from for context.

3. Exposure (appendix) “While the majority of them was receiving the benefits until the end of the cohort follow-up, some individuals ended their receipt before, but still we considered the end of the cohort follow-up as end of their exposure period. This was because individuals that improve their economic conditions -and are not eligible anymore for PBF- should exit the PBF” — this is a defensible approach, but people could also lose PBF for not completing conditionalities, programmatic reasons, challenges with access (the authors argue this is relatively uncommon). Consider a sensitivity analysis whereby these individuals are either censored or put back into the unexposed category after loss of benefits.

4. Table 1 convincingly shows that beneficiaries and non-beneficiaries are quite different, showing the need for an approach like IPTW. It would be useful to show a similar table after weighting is applied, demonstrating if/that the re-weighted population now appears similar.

5. I’m not sure that I would have chosen a Poisson regression as the primary approach, vs a survival analysis, but it is reassuring to see similar results in the triangulation analysis. Was IPTW used for this alternative approach as well? Please indicate this in the description in the appendix.

6. Case fatality outcome – ART adherence is almost certainly a mediator between cash transfers and death, and therefore I recommend against adjusting for this in the primary model.

7. Covariates — there are likely program-related factors that vary at the municipal level and impact program access (e.g., outreach, case workers, logistics, etc). Is it possible to control for these in the model (perhaps by including the participation rate standardized to the amount of poverty?)

8. It would be useful to include temporal analyses based on the duration of PBF exposure. In other words, does protection against AIDS etc grow over time? In our paper (Richterman and Thirumurthy, Nature Human Behavior 2022), we showed that AIDS-related deaths decrease over time after cash transfer program implementation at the population-level, but could not differentiate between whether this was related to general program expansion over time (i.e. more people receiving the benefit) OR greater length of time exposed for individuals. This is an important gap in the literature that this analysis is well-positioned to address.

9. Is there substantial variation in benefit size and is this captured well by your datasets? An alternative explanation for the gradient of effect by wealth would be that poorer individuals receive larger benefits, and it is the size of the benefit that is in part driving this difference. If possible this would be worth exploring by looking at sub-groups based on benefit size.

Reviewer #3 (Remarks to the Author):

"The impact of conditional cash transfers on AIDS incidence, mortality, and case-fatality is determined by poverty levels: a cohort study of 22.7 million individuals in Brazil" by Professor Rasella

What are the noteworthy results?

This study is a whole of population analysis of the conditional cash transfers (CCTs) as conducted in Brazil through Programa Bolsa Familia (PBF) on the incidence of AIDS, AIDS mortality and case fatality rate. The study demonstrated substantial and statistically significant protective effect across these outcomes with additional benefit for those in the lowest quartile of wealth and adolescents and young people. The substantial protective effect of delivery of a socially policy that particularly targets those most at risk of poor outcomes is significant on a global scale. The choice to analyse impact on incidence of AIDS also demonstrates the ability of this intervention to reduce outcomes with significant health and social determinants including stigma and discrimination; which further contribute to the impact of poverty.

- Will the work be of significance to the field and related fields? How does it compare to the established literature? If the work is not original, please provide relevant references.

This is a highly original piece of work. This is the largest analysis of a population cohort exposed to a CCT demonstrating effectiveness against AIDS. Poverty alleviation interventions have been demonstrated to have substantial impact on social outcomes. This analysis adds to the understanding of the breadth of impact of programs of this type. It demonstrates that positive outcomes can be achieved in a heterogenous population and particularly for those with the least wealth, for whom these policies are targeted towards.

- Does the work support the conclusions and claims, or is additional evidence needed?

The primary outcomes of the analysis are robust and largely support the conclusions of the study. See next two points for elaboration.

Considering the demographics of the HIV/AIDS epidemic and poverty in Brazil, appropriate consideration has been given to MSM and sex work. Within this context it would also be worth considering the effectiveness of PBF among people who have been incarcerated and/or use illicit drugs. While the data collection limitations may mean that stratified analyses cannot be conducted for these populations, it would be good to at least consider implications for these populations in the discussion.

- Are there any flaws in the data analysis, interpretation and conclusions? - Do these prohibit publication or require revision?

The statement that PBF benefits were stronger for women, lines 47 and 192, is not aligned with the data presented in Table 3. The estimates and 95% CIs in this table do not demonstrate a significant or meaningful difference in PBF effect on between women and men on AIDS incidence (IRR 0.60, 95%CI 0.44-0.51; IRR 0.61, 95%CI 0.57-0.63 respectively); similarly for mortality and case fatality (substantially overlapping confidence intervals). That impacts in women are as substantial as in men is of importance in the context of reduction in HIV/AIDS.

The interpretation of the stratified analyses could be enhanced by including a test for interaction. These issues do not prohibit publication if satisfactorily addressed in the text.

- Is the methodology sound? Does the work meet the expected standards in your field?

The authors have undertaken substantial and extensive sensitivity analyses to assess the robustness outcomes. This was necessary and appropriate considering the observational/quasi experimental nature of the study. These analyses considered primary potential limitations of the design including adjustments for geographical/municipal level characteristics which were necessary considering the geographic differential of risk of HIV infection across Brazil, and exploring estimates of wealth – a key parameter of this analysis. A variety of modelling strategies and evaluation of missing data are also considered. That the primary associations are consistent across the range of analyses is a strength of this study and support the robustness of claims for this quasi experimental design.

- Is there enough detail provided in the methods for the work to be reproduced?

The statistical methods and study details are comprehensive and well described. I would suggest review by a statistical reviewer with expertise in the methods used.

REBUTTAL LETTER

Response to Reviewers

Reviewer #1 (Remarks to the Author):

Question(Q)1.1: Overall this is an impressive analysis of a very large cohort and the results of the PBF on AIDS outcomes are large. Very important findings. That said, there are some issues with the cohort construction and variable description that if better clarified could strengthen the paper. Overall being more clear about eligibility for the PBF in the paper (not just linking to a citation), being more clear in main text about entry/eligibility into the primary data set (from which it seems health metrics were linked) and being more clear about metrics (e.g. HIV incidence?) - these will significantly strengthen the paper.

A1.1: We thank the reviewer for acknowledging the high relevance of our study and the completeness of the analyses carried out.

As suggested, we have improved the clarity of some explanations in the main text.

Regarding the description of the cohort construction, while it is explained in detail in the Supplementary Appendix, on page 3, we have added in the methods section of the main text, p 16-17 (lines 551-583), an entire paragraph with a description of its construction: “The 100 Million Brazilian Cohort was based on the baseline information of families, during the period from January 1, 2001 to December 31, 2017, who sought to benefit from the Brazilian government's social programs through registration in the Unified Registry for Social Programs (in Portuguese: *Cadastro Único para Programas Sociais – Cadastro Único*). The Unified Registry is an administrative database, to which Brazilians aged 16 or over can apply by registering their personal information (age, sex, race, education and others) and household information (household density, familiar income, structural characteristics of the residence and others), as long as they are within one of these categories: (i) belong to a family with a monthly per capita income of up to half a minimum wage; (ii) belong to a family with a total monthly income of up to three minimum wages; (iii) belong to a family with an income greater than three minimum wages, provided that the registration is linked to inclusion in social programs in the three spheres of government; (iv) be the only resident of the household, or; (v) living on the streets (alone or with the family). Upon registration, individuals receive a unique identifier code and are searched for socioeconomic characteristics. At the end of 2017, Unified Registry had approximately 114 million individuals on its register, which represents around 50% of the Brazilian population. It is a social tool that identifies and characterizes especially low-income families, allowing the government to know the socioeconomic aspect of the poorest and use it for the selection of social programs. Two individual-level health-related datasets were linked to the Unified Registry for Social Programs: the Notifiable Diseases Information System (SINAN) and the Mortality Information System (SIM). SINAN is a decentralized information system that monitors the incidence of notifiable diseases, including HIV/AIDS. SIM is a national death surveillance system that registers deaths by all causes, including HIV/AIDS, according to CID-10 classification. Created by the Center for Integration of Data and Knowledge for Health (CIDACS / FIOCRUZ), the Cohort aims to facilitate research and continuous assessment of social determinants and the effects of social policies and programs in health contexts in Brazil. It has 246 variables with demographic and socioeconomic information at the individual and family level. The codes and linking algorithms between the databases were built to make efficient and specific links through five identifiers: the date of birth, the municipality of residence, the sex, the name and the mother's name of each individual presented in each of the databases. The linking codes and algorithms were built based on five identifiers: date of birth, place of residence, sex, name, and mother's name of the individual present on each database. The Unified Registry and the health datasets (SIM and SINAN) were individually matched in two steps, using the CIDACS-RL tool (appendix,p.3).^{20,23} The quality of each link between Unified Registry, SINAN e SIM has been extensively evaluated and validated.”

Regarding the eligibility for the PBF in the paper, we have added to the explanation in the Methods section, page 17, line 604-611, the following paragraph: “The beneficiary group was defined as eligible individuals who received PBF benefits [extremely poor families are considered to have a monthly per capita income of up to US\$12 (BRL 60) in 2007-2008 period, up to US\$14 (BRL 70) from 2009 to 2013, and up to US\$15.4 (BRL 77) in 2014 and 2015. Poor families are those with a per capita income between US\$12 (BRL 60.01) and US\$24 (BRL 120) in 2007 and 2008, between US\$14 (BRL 70.01) and US\$28 (BRL 140) from 2009 to 2013, and between US\$15.4 (BRL 77.01) and US\$30.8 (BRL 154) in 2014 and 2015], and their exposure started with the receipt of the benefit, until the end of their cohort follow-up.”

The description of the variable metrics has also been improved in Methods section, page 17, line 595-597, adding: "The AIDS outcomes available in the final dataset were: (i) new AIDS cases, defined by adapted CDC criteria, the Rio de Janeiro/Caracas criteria; and (ii) AIDS deaths, considering as underlying cause the ICD-10 codes B20 to B24."

Q1.2: Please better describe who the comparison group is in this analysis- those who didn't get PBF but if the primary data set is everyone in the social protection data set- why would they not have been eligible? Just be more clear about study design and comparator in the main text.

A1.2: The description of the comparison group has been improved in the Methods section, page 18, line 611-617, and complemented with a new description about eligible individuals that were classified in the non-beneficiary group: "The non-beneficiary group was defined as individuals who had never benefited from PBF throughout their follow-up period, as in previous studies. In case of administrative delays and non-receipt of the benefits, eligible individuals were classified in the non-beneficiary group (details provided in p.4). Administrative delays occur because after registering with *Cadastro Único* and fulfilling the aforementioned requirements, the Ministry of Citizenship (in Portuguese: *Ministério da Cidadania*) elects the beneficiaries of the program by following the previously described criteria."

Q1.3: While the authors cite original publications for the 100 million Brazilian cohort- how was that actually constructed? It would help the reader to briefly say something on how individuals entered into that cohort (how were they identified, eligibility) Is everyone in it eligible for the PBF? This was not totally clear given the statement that it was created from the unified registry for social protection (line 136).

A1.3: We included the explanation of how the Cohort was constructed in the Methods section, page 16-17, line 551-588: "The 100 Million Brazilian Cohort was based on the baseline information of families, during the period from January 1, 2001 to December 31, 2017, who sought to benefit from the Brazilian government's social programs through registration in the Unified Registry for Social Programs (in Portuguese: *Cadastro Único para Programas Sociais – Cadastro Único*). The Unified Registry is an administrative database, to which Brazilians aged 16 or over can apply by registering their personal information (age, sex, race, education and others) and household information (household density, familiar income, structural characteristics of the residence and others), as long as they are within one of these categories: (i) belong to a family with a monthly per capita income of up to half a minimum wage; (ii) belong to a family with a total monthly income of up to three minimum wages; (iii) belong to a family with an income greater than three minimum wages, provided that the registration is linked to inclusion in social programs in the three spheres of government; (iv) be the only resident of the household, or; (v) living on the streets (alone or with the family). Upon registration, individuals receive a unique identifier code and are searched for socioeconomic characteristics. At the end of 2017, Unified Registry had approximately 114 million individuals on its register, which represents around 50% of the Brazilian population. It is a social tool that identifies and characterizes especially low-income families, allowing the government to know the socioeconomic aspect of the poorest and use it for the selection of social programs. Two individual-level health-related datasets were linked to the Unified Registry for Social Programs: the Notifiable Diseases Information System (SINAN) and the Mortality Information System (SIM). SINAN is a decentralized information system that monitors the incidence of notifiable diseases, including HIV/AIDS. SIM is a national death surveillance system that registers deaths by all causes, including HIV/AIDS, according to CID-10 classification. Created by the Center for Integration of Data and Knowledge for Health (CIDACS / FIOCRUZ), the Cohort aims to facilitate research and continuous assessment of social determinants and the effects of social policies and programs in health contexts in Brazil. It has 246 variables with demographic and socioeconomic information at the individual and family level. The codes and linking algorithms between the databases were built to make efficient and specific links through five identifiers: the date of birth, the municipality of residence, the sex, the name and the mother's name of each individual presented in each of the databases. The linking codes and algorithms were built based on five identifiers: date of birth, place of residence, sex, name, and mother's name of the individual present on each database. The Unified Registry and the health datasets (SIM and SINAN) were individually matched in two steps, using the CIDACS-RL tool (appendix,p.3).^{20,23} The quality of each link between Unified Registry, SINAN e SIM has been extensively evaluated and validated."

Not all individuals registered in *Cadastro Unico* are eligible for PBF, as we state in the same subsection at p.17-18, line 604-611: "The beneficiary group was defined as eligible individuals who received PBF benefits [extremely poor families are considered to have a monthly per capita income of up to US\$12 (BRL 60) in 2007-2008 period, up to US\$14 (BRL 70) from 2009 to 2013, and up to US\$15.4 (BRL 77) in 2014 and 2015. Poor families are those with a per capita income between US\$12 (BRL 60.01) and US\$24 (BRL 120) in 2007 and 2008, between US\$14 (BRL 70.01) and US\$28 (BRL 140) from 2009 to 2013, and between US\$15.4 (BRL 77.01) and US\$30.8 (BRL 154) in 2014 and 2015], and their exposure started with the receipt of the benefit, until the end of their cohort follow-up."

Q1.4. Also just stating in the text the eligibility criteria for the PBF would also help in understanding who is and is not in the data set- the citation included (Rasella Lancet 2013) seems to suggest the program is for families that have children or pregnant women_so if this is the case it would suggest the target population for this analysis is more focused than suggested. Please state who is eligible.

A1.4: These eligibility criteria have been added in the text, in the Methods section, page 17 -18, line 604-611: “The beneficiary group was defined as eligible individuals who received PBF benefits [extremely poor families are considered to have a monthly per capita income of up to US\$12 (BRL 60) in 2007-2008 period, up to US\$14 (BRL 70) from 2009 to 2013, and up to US\$15.4 (BRL 77) in 2014 and 2015. Poor families are those with a per capita income between US\$12 (BRL 60.01) and US\$24 (BRL 120) in 2007 and 2008, between US\$14 (BRL 70.01) and US\$28 (BRL 140) from 2009 to 2013, and between US\$15.4 (BRL 77.01) and US\$30.8 (BRL 154) in 2014 and 2015], and their exposure started with the receipt of the benefit, until the end of their cohort follow-up.”

A1.5: Please clarify if HIV incidence was measured and how? I see new AIDS diagnoses which I understand it being called AIDS incidence- would prefer it just be called new AIDS diagnoses.

Q1.5: We thank the reviewer for the remark, we considered HIV notifications as proxies of HIV incidence and for this reason we named HIV incidence all over the manuscript. However, in the case of HIV, we recognize that the potential not-negligible HIV under notification could indicate to use HIV notification instead. For this reason, we have changed the term HIV incidence with HIV notification rate all over the text. Regarding AIDS, we would respectfully prefer to maintain the term AIDS incidence over the text, as we already did in previous publications with these data^{1,2} because we consider that in the Brazilian context, considering the high detection rate of the notification system, AIDS notifications could be considered proxies of the real incidence, as considered in the official reports of the Ministry of Health.³

Q1.6: To the point about HIV incidence “HIV cases notified in the database after 2013, and we found that receiving PBF benefits was associated to a 36% decrease in the HIV incidence rate (RR: 0.64; 95% CI: 0.62-0.66) (Table S28).” Its not clear this is HIV incidence, again it is new HIV diagnoses (yes...new? Not sure HIV diagnoses) but it doesn't mean the infection is a new infection does it? It could be the person has been infected for years but was just diagnosed. Unless using a test that captures incidence? Seems not. Again use language that more accurately represents measure (HIV tests?)

A1.6: We agree with the reviewer and, as already mentioned in the previous response, we have changed the term HIV incidence with HIV notification rate all over the text.

Q1.7: Line 215 “proportion of Men who have Sex with Men (MSM) within the PLWA cohort was relatively low, accounting for only 15.0% of the 26,936 PLWA individuals, indicating that HIV transmission within this cohort of economically disadvantaged Brazilians was primarily due to heterosexual relationships.” I would be careful about this statement- again it depends on the sampling frame and who was entered into the original data set for linkage- it if is individuals eligible for PBF is there some bias in who is in the dataset (seem likely since many new HIV infections in Brazil are among MSM)- raising questions again as to whether social protections are reaching key populations- I am not sure this data set can answer this question since there is selection bias (yes?) in who is in the original data?

A1.7: We agree with the reviewer that many cases of HIV/AIDS in Brazil are among risk populations, in particular MSM. However, it has to be considered that in the overall population still around 50% of new AIDS cases in the last decade have been notified among heterosexual individuals.³ While we recognize that we cannot infer, from the percentage of MSM in our PLWA cohort, the percentage of MSM in the all study cohort, it was important to evaluate using the available data in these most vulnerable individuals sexual risk behaviors related to poverty, such as exchange sex and prostitution, which are usually mainly related to an heterosexual transmission, could be the among the main drivers of the transmission.

However, we have modified the sentence in line 218-221 as: “proportion of Men who have Sex with Men (MSM) within the PLWA cohort was accounting for 15.0% of the 26,936 PLWA individuals, suggesting that HIV transmission within this cohort of economically disadvantaged Brazilians could be in a large part due to heterosexual relationships.”

Reviewer #2 (Remarks to the Author):

Authors report an analysis assessing the causal effects of a large conditional cash transfer program in Brazil (Programa Bolsa Família - PBF) on several important HIV-related outcomes, finding reductions in AIDS incidence, AIDS-related mortality, and AIDS case-fatality. They also find suggestive evidence that PBF leads to reductions in HIV incidence. Causal claims are further supported by a gradient effect by wealth quintile, whereby the strongest effects are seen among the poorest participants. This is a very important, innovative, and unique study that is able to connect several very large datasets and assess the effects of an exemplary conditional cash transfer program on key HIV outcomes. I generally found the methodology to be of the highest standard, with extensive sensitivity analyses addressing many of my initial concerns. This work adds to the evidence that social protection and poverty reduction programs are going to be critical components of efforts to end HIV epidemics now that we are in an era of widespread access to antiretroviral therapy. I have a few suggestions for the authors to consider that I believe would strengthen this work.

A2.0: We thank the reviewer for recognizing the innovativeness and uniqueness of our study, and the high standard of the analyses complemented by a wide range of sensitivity tests. We have enhanced the clarity and robustness of the paper thanks to your constructive suggestions.

Q2.1. Some references are a bit outdated in the introduction. Consider putting in the context of more recent systematic reviews of cash transfer effects on general outcomes (Bestagli et al, ODI 2016) and HIV outcomes (Guimarães et al, Lancet HIV 2023 — I believe co-authored by some of the authors of this manuscript), and our recent paper directly evaluating cash transfer effects on HIV-related outcomes in 42 LMICs (Richterman and Thirumurthy, Nature Human Behaviour 2022).

A2.1: We thank the reviewer for suggesting to update some references, we have included in the introduction, page 3, line 96 the references by Guimarães et al. (Lancet HIV, 2023) and Richterman and Thirumurthy (Nature Human Behavior, 2022).

Q2.2. Line 102 — I imagine the income limits and benefits listed have not been static over the 20 years program history, are these figures from a specific year? Please specify what year they come from for context.

A2.2: The sentence "defined as families earning between US\$18-36 per person per month" refers to the year 2018 mentioned above. We have also inserted it after the information, on line 104, page 3.

Q2.3. Exposure (appendix) "While the majority of them was receiving the benefits until the end of the cohort follow-up, some individuals ended their receipt before, but still we considered the end of the cohort follow-up as end of their exposure period. This was because individuals that improve their economic conditions -and are not eligible anymore for PBF- should exit the PBF" — this is a defensible approach, but people could also lose PBF for not completing conditionalities, programmatic reasons, challenges with access (the authors argue this is relatively uncommon). Consider a sensitivity analysis whereby these individuals are either censored or put back into the unexposed category after loss of benefits.

A2.3: For the beneficiaries, we have decided to define as exposure time also the period after the end of the receipt of the benefits in order to capture more comprehensively the effects of PBF. While we have no data to discriminate which individuals finished receiving the benefits because of the increase of their income per capita above the eligibility thresholds, or because of the non-compliance of program conditionalities, reports show that the second category is a negligible part in comparison to the first.⁴ Considering that almost the totality of individuals stops receiving PBF benefits because of the improvement of their living conditions, and that this improvement, as studies show,⁵⁻⁷ is due in the large majority of the cases to the PBF cash transfers and PBF conditionalities, we consider the reduced risk to evolve to AIDS or to die from AIDS of these individuals as an effect of the previous exposure to PBF.

Conditional Cash Transfers are massive poverty-reduction policies, and PBF impact have been evaluated in terms of long-term effects⁸ and also in terms of spill-over effects in the community.⁹

In order to estimate how much of this PBF effects was exerted during the receipt of benefits, we have conducted sensitivity analyses (see Table S29 in the webappendix), creating a new group of PBF beneficiaries, with their follow-up time ending on the date of the termination of cash transfer receipt. As it can be seen, in these exposed individuals the PBF effects has the same direction and similar magnitude of the main results. However, considering all the

reasons listed above, we prefer to maintain as follow-up time for the exposed group the one presented in the main text.

Q2.4. Table 1 convincingly shows that beneficiaries and non-beneficiaries are quite different, showing the need for an approach like IPTW. It would be useful to show a similar table after weighting is applied, demonstrating if/that the re-weighted population now appears similar.

A2.4. We added a similar table in the appendix (Table S31, page 46), demonstrating that the behavior of covariates between the PBF and N-PBF groups after applying IPTW weighting is similar to the one without IPTW. We also include a citation to this table in the manuscript, page 5, lines 161-163: "The appendix contains additional analysis (Table S31, page 46) that demonstrates the behavior of covariates between the beneficiaries and non-beneficiaries groups after applying IPTW weighting."

Q2.5. I'm not sure that I would have chosen a Poisson regression as the primary approach, vs a survival analysis, but it is reassuring to see similar results in the triangulation analysis. Was IPTW used for this alternative approach as well? Please indicate this in the description in the appendix.

A2.5. We confirm that the Inverse Probability of Treatment Weight (IPTW) was used also in the survival analysis. We also included in the appendix description, in "5.Triangulation Analyses", page 34, the sentence: "The results estimated using the survival models were weighted by the Inverse Probability of Treatment Weight (IPTW)."

Q2.6. Case fatality outcome – ART adherence is almost certainly a mediator between cash transfers and death, and therefore I recommend against adjusting for this in the primary model.

A2.6. We appreciate the suggestion, and we included a Poisson model without ART adherence as a supplementary analysis in the appendix (Table S30, page 45). However, we did not observe significant changes in the effect of the PBF, as in the model with ART the PBF effect was 25% (RR: 0.75;95%CI: 0.66-0.85), and without ART it was 24% (RR: 0.76; 95%CI: 0.67-0.86).

We recognize that ART can be seen as mediator between cash transfers and death from AIDS, but we believe that it could also be considered a confounding factor, because it could enable and facilitate the PBF enrollment, receipt of PBF benefits, and compliance with PBF conditionalities, among others. For this reason, we would respectfully prefer to maintain ART in the final model presented in the main text, if possible. To better clarify this issue, we also included in the main text at p 20, line 690-693: "Due to the ART potential mediator effects in the relationship between PBF coverage and mortality from AIDS, we have also fitted the case-fatality rate models without ART as independent variable, with no changes in the PBF estimates (see appendix, Table S30, page 45)."

Q2.7. Covariates — there are likely program-related factors that vary at the municipal level and impact program access (e.g., outreach, case workers, logistics, etc). Is it possible to control for these in the model (perhaps by including the participation rate standardized to the amount of poverty?)

A2.7. Thank you for the suggestion, for the reason cited above we have already incorporated a wide range of available municipal-level variables (as presented in the main models in Table 2) that control for infrastructure in municipalities (such as inadequate sanitation), economic dynamics (such as the unemployment rate), and a range of healthcare service-related variables. In addition, we have included several sensitivity tests with other municipal-level variables in Tables S11, S12 and S13 of the appendix, and the municipal-level variables added have been percentage of the municipal population with garbage collection, GDP per capita, proportion of poor people, rate of specialized clinics per 1,000 inhabitants, per capita income, proportion of extremely poor population and the municipal Gini index. None of these variables, once included, has significantly changed the association between PBF coverage the health outcomes.

Q2.8. It would be useful to include temporal analyses based on the duration of PBF exposure. In other words, does protection against AIDS etc grow over time? In our paper (Richterman and Thirumurthy, Nature Human Behavior 2022), we showed that AIDS-related deaths decrease over time after cash transfer program implementation at the population-level, but could not differentiate between whether this was related to general program expansion over time (i.e. more people receiving the benefit) OR greater length of time exposed for individuals. This is an important gap in the literature that this analysis is well-positioned to address.

A2.8. We agree on the existing gap in the literature and on the importance of evaluating temporal analyses. We thank the reviewer for the suggestion, however, given the already extensive scope of our analysis, and the need to properly and comprehensively evaluate this and other specific Cash Transfer modulators of effectiveness, we feel that such

analysis would be out of scope in this manuscript. We are currently developing another study with the same dataset where we are evaluating all these aspects, including temporal dimensions of effectiveness.

Q2.9. Is there substantial variation in benefit size and is this captured well by your datasets? An alternative explanation for the gradient of effect by wealth would be that poorer individuals receive larger benefits, and it is the size of the benefit that is in part driving this difference. If possible this would be worth exploring by looking at sub-groups based on benefit size.

A2.9. We thank the reviewer for the suggestion, that we consider highly relevant for the study. However, at the moment, we do not have information on the amount of the benefits in our available dataset. We are currently working on a specific linkage with the national PBF payroll system to obtain this information, and perform such analysis.

Reviewer #3 (Remarks to the Author):

"The impact of conditional cash transfers on AIDS incidence, mortality, and case-fatality is determined by poverty levels: a cohort study of 22.7 million individuals in Brazil" by Professor Rasella

What are the noteworthy results?

This study is a whole of population analysis of the conditional cash transfers (CCTs) as conducted in Brazil through Programa Bolsa Familia (PBF) on the incidence of AIDS, AIDS mortality and case fatality rate. The study demonstrated substantial and statistically significant protective effect across these outcomes with additional benefit for those in the lowest quartile of wealth and adolescents and young people. The substantial protective effect of delivery of a socially policy that particularly targets those most at risk of poor outcomes is significant on a global scale. The choice to analyse impact on incidence of AIDS also demonstrates the ability of this intervention to reduce outcomes with significant health and social determinants including stigma and discrimination; which further contribute to the impact of poverty.

- Will the work be of significance to the field and related fields? How does it compare to the established literature? If the work is not original, please provide relevant references.

This is a highly original piece of work. This is the largest analysis of a population cohort exposed to a CCT demonstrating effectiveness against AIDS. Poverty alleviation interventions have been demonstrated to have substantial impact on social outcomes. This analysis adds to the understanding of the breadth of impact of programs of this type. It demonstrates that positive outcomes can be achieved in a heterogenous population and particularly for those with the least wealth, for whom these policies are targeted towards.

- Does the work support the conclusions and claims, or is additional evidence needed?

The primary outcomes of the analysis are robust and largely support the conclusions of the study. See next two points for elaboration.

Q1. Considering the demographics of the HIV/AIDS epidemic and poverty in Brazil, appropriate consideration has been given to MSM and sex work. Within this context it would also be worth considering the effectiveness of PBF among people who have been incarcerated and/or use illicit drugs. While the data collection limitations may mean that stratified analyses cannot be conducted for these populations, it would be good to at least consider implications for these populations in the discussion.

A3.1. We greatly appreciate the reviewer's acknowledgment of the robustness of our analyses. Indeed, we considered the possibility of analyzing more subpopulations, such as people who have been incarcerated and/or use illicit drugs. However, these variables are not available in the database or do not have an adequate quality, as a consequence stratified analyses are not possible. However, we recognize the importance of considering the implications for these population, and we added in the results section, p. 7, line 231-235: "While further stratification analyses of extremely vulnerable populations, such as incarcerated individuals and users of illicit drugs, have not been possible due to the lack of these variables in the study cohort, it is plausible that PBF could exert a particularly strong effect on these populations, at least of similar magnitude what have been shown in extremely poor individuals."

- Are there any flaws in the data analysis, interpretation and conclusions? - Do these prohibit publication or require revision?

The statement that PBF benefits were stronger for women, lines 47 and 192, is not aligned with the data presented in Table 3. The estimates and 95% CIs in this table do not demonstrate a significant or meaningful difference in PBF effect on between women and men on AIDS incidence (IRR 0.60, 95%CI 0.44-0.51; IRR 0.61, 95%CI 0.57-0.63 respectively); similarly for mortality and case fatality (substantially overlapping confidence intervals). That impacts in women are as substantial as in men is of importance in the context of reduction in HIV/AIDS. The interpretation of the stratified analyses could be enhanced by including a test for interaction. These issues do not prohibit publication if satisfactorily addressed in the text.

A3.2: We agree with the reviewer that, even of there are some differences in the point estimates, they have substantially overlapping intervals. To further verify any effect modification, we performed an interaction analysis as

suggested by the reviewer: in Table S32, page 47, we show that the interaction term added to the multivariable model for the association between PBF exposure and AIDS incidence is not statistically significant. For this reason, in the text we have removed any mention of the PBF effect being more pronounced in women compared to men.

- Is the methodology sound? Does the work meet the expected standards in your field?

The authors have undertaken substantial and extensive sensitivity analyses to assess the robustness outcomes. This was necessary and appropriate considering the observational/quasi experimental nature of the study. These analyses considered primary potential limitations of the design including adjustments for geographical/municipal level characteristics which were necessary considering the geographic differential of risk of HIV infection across Brazil, and exploring estimates of wealth – a key parameter of this analysis. A variety of modelling strategies and evaluation of missing data are also considered. That the primary associations are consistent across the range of analyses is a strength of this study and support the robustness of claims for this quasi experimental design.

A3.3: We thank the reviewer for the acknowledgment of the robustness of the study and of the extensiveness of the sensitivity and triangulation analyses.

- Is there enough detail provided in the methods for the work to be reproduced?

The statistical methods and study details are comprehensive and well described. I would suggest review by a statistical reviewer with expertise in the methods used.

REFERENCES

1. Morais, G. A. de S. et al. Effect of a conditional cash transfer programme on AIDS incidence, hospitalisations, and mortality in Brazil: a longitudinal ecological study. *Lancet HIV* 9, e690–e699 (2022).
2. Lua, I. et al. The effects of social determinants of health on acquired immune deficiency syndrome in a low-income population of Brazil: a retrospective cohort study of 28.3 million individuals. www.thelancet.com (2023).
3. BRASIL- Saúde, M. da. Boletim Epidemiológico HIV / Aids | 2022. Secretaria de Vigilância em Saúde (2022).
4. Roma, J. et al. 7-8 15-16 29-30 Boletim Bolsa Família e Cadastro Único No72 2. www.cidadania.gov.br.
5. Campello T (Organização), N. M. (Organização). BOLSA FAMÍLIA PROGRAMA BOLSA FAMÍLIA-UMA DÉCADA DE INCLUSÃO E CIDADANIA. www.ipea.gov.br. 2013 [cited 16 May 2023].
6. Mourão, L. & Jesus, A. M. de. Bolsa Família (Family Grant) Programme: an analysis of Brazilian income transfer programme. <http://journals.openedition.org/factsreports> (2011).
7. Lindert, K., Linder, A., Hobbs, J. & De La Brière, B. The Nuts and Bolts of Brazil's Bolsa Família Program: Implementing Conditional Cash Transfers in a Decentralized Context. (2007).
8. Rasella, D. et al. Long-term impact of a conditional cash transfer programme on maternal mortality: a nationwide analysis of Brazilian longitudinal data. *BMC Med* 19, 1–9 (2021).
9. Rasella, D., Aquino, R., Santos, C. A. T., Paes-Sousa, R. & Barreto, M. L. Effect of a conditional cash transfer programme on childhood mortality: A nationwide analysis of Brazilian municipalities. *The Lancet* 382, 57–64 (2013).

REVIEWERS' COMMENTS

Reviewer #1 (Remarks to the Author):

Overall the authors have been responsive to the comments.

I am still a bit confused about the detailed description of entry into the Unified Registry for Social Programs and how it relates to eligibility for the PBF.

The information provided on entry into the social registry:

(i) belong to a family with a monthly per capita income of up to half a minimum wage; (ii) belong to a family with a total monthly income of up to three minimum wages; (iii) belong to a family with an income greater than three minimum wages, provided that the registration is linked to inclusion in social programs in the three spheres of government; (iv) be the only resident of the household, or; (v) living on the streets (alone or with the family).

How do these criteria relate to:

"The beneficiary group was defined as eligible individuals who received PBF benefits [extremely poor families are considered to have a monthly per capita income of up to US\$12 (BRL 60) in 2007-2008 period, up to US\$14 (BRL 70) from 2009 to 2013, and up to US\$15.4 (BRL 77) in 2014 and 2015. Poor families are those with a per capita income between US\$12 (BRL 60.01) and US\$24 (BRL 120) in 2007 and 2008, between US\$14 (BRL 70.01) and US\$28 (BRL 140) from 2009 to 2013, and between US\$15.4 (BRL 77.01) and US\$30.8 (BRL 154) in 2014 and 2015]"

For example, how do these monthly per capita incomes relate to the registry 'half a minimum wage' etc.

Just to be clear- it seems everyone in the registry should be eligible for the PBF but not everyone received benefits for various reasons and they are the comparison group? Is that so?

From the comparison of the two groups the non-beneficiary group is better off on many SES variables which the IPTW balances...which is important.

Reviewer #2 (Remarks to the Author):

I thank the authors for their detailed responses and revisions, which fully address my concerns.

The one point I would make is to strongly encourage pursuit of temporal analyses with these datasets, as understanding the effects of CTs over time is highly policy relevant. I understand this may be beyond the scope of this particular manuscript (which I agree is quite lengthy already), and defer to the authors and editors whether this is the case.

REBUTTAL LETTER

Response to Reviewers

Reviewer #1 (Remarks to the Author):

Question(Q)1.1: Overall the authors have been responsive to the comments.

I am still a bit confused about the detailed description of entry into the Unified Registry for Social Programs and how to relates to eligibility for the PBF.

The information provided on entry into the social registry:

(i) belong to a family with a monthly per capita income of up to half a minimum wage; (ii) belong to a family with a total monthly income of up to three minimum wages; (iii) belong to a family with an income greater than three minimum wages, provided that the registration is linked to inclusion in social programs in the three spheres of government; (iv) be the only resident of the household, or; (v) living on the streets (alone or with the family).

How do these criteria relate to:

The beneficiary group was defined as eligible individuals who received PBF benefits [extremely poor families are considered to have a monthly per capita income of up to US\$12 (BRL 60) in 2007-2008 period, up to US\$14 (BRL 70) from 2009 to 2013, and up to US\$15.4 (BRL 77) in 2014 and 2015. Poor families are those with a per capita income between US\$12 (BRL 60.01) and US\$24 (BRL 120) in 2007 and 2008, between US\$14 (BRL 70.01) and US\$28 (BRL 140) from 2009 to 2013, and between US\$15.4 (BRL 77.01) and US\$30.8 (BRL 154) in 2014 and 2015]" For example, how do these monthly per capita incomes relate to the registry 'half a minimum wage' etc.

Just to be clear- it seems everyone in the registry should be eligible for the PBF but not everyone received benefits for various reasons and they are the comparison group? Is that so?

From the comparison of the two groups the non-beneficiary group is better off on many SES variables which the IPTW balances...which is important.

A1.1: The registration on the CadÚnico is a prerequisite for receiving the PBF (or any other federal social protection program), but it does not imply approval or immediate receipt. After registering with CadÚnico and fulfilling the aforementioned requirements, the Ministry of Citizenship (in Portuguese: Ministério da Cidadania) elects the beneficiaries of the program by following the previously described criteria. Once the income transfer is approved, there are conditions for its maintenance according to each family composition.

The beneficiary group was defined as the PBF eligible individuals according to the a monthly per capita income (previously described). Which is within the category '(i) belong to a family with a monthly per capita income of up to half a minimum wage' of the Unified Registry for Social Programs.

The non-beneficiary group was defined as individuals who had never benefited from PBF throughout their follow-up period, as in previous studies. In case of administrative delays and non-receipt of the benefits, eligible individuals were classified in the non-beneficiary group. Administrative delays occur because after registering with Cadastro Único and fulfilling the aforementioned requirements, the Ministry of Citizenship elects the beneficiaries of the program by following the previously described criteria.

Reviewer #2 (Remarks to the Author):

Q2.1. I thank the authors for their detailed responses and revisions, which fully address my concerns.

The one point I would make is to strongly encourage pursuit of temporal analyses with these datasets, as understanding the effects of CTs over time is highly policy relevant. I understand this may be beyond the scope of this particular manuscript (which I agree is quite lengthy already), and defer to the authors and editors whether this is the case.

A2.1: We thank the reviewer for recognizing the innovativeness and uniqueness of our study, and suggestions for studies with temporal analysis. We will take it into consideration.